# INVERTING DATA TRANSFORMATIONS VIA DIFFUSION SAMPLING

## ABSTRACT

We study the problem of *transformation inversion* on general Lie groups: a datum is transformed by an unknown group element, and the goal is to recover an inverse transformation that maps it back to the original data distribution. Such unknown transformations arise widely in machine learning and scientific modeling, where they can significantly distort observations. As a key application, we focus on *test-time equivariance*, where the objective is to improve the robustness of pretrained neural networks to input transformations at inference time and without any (re)training. We take a probabilistic view and model the posterior over transformations as a Boltzmann distribution defined by an energy function in data space. To sample from this posterior, we introduce a diffusion process on Lie groups that keeps all updates on-manifold and only requires computations in the associated Lie algebra. Our method, *Transformation-Inverting Energy Diffusion* (TIED), relies on a new trivialized target-score identity that enables efficient score-based sampling of the transformation posterior. TIED naturally handles curved group geometries and rugged, multimodal energy landscapes, and it applies to a broad class of Lie groups and nonlinear actions without assuming compactness or a bi-invariant metric. Experiments on image homographies and PDE symmetries demonstrate that TIED can restore transformed inputs to the training distribution at test time, showing improved performance over strong canonicalization and sampling baselines.

## 1 INTRODUCTION

Many data modalities are observed after unknown or unusual geometric transformations. In this paper, we ask a general question: given data transformed by an unknown element of a Lie group, can we recover an inverse transformation that returns it to the original distribution? This problem appears in many scenarios, including viewpoint changes and projective warps in computer vision, sensor motion and registration in medical imaging, and changes of reference frame in scientific modeling (Olver, 1993; Hartley & Zisserman, 2003; Celledoni et al., 2021). It is an instance of a blind inverse problem since the nuisance transform is unknown and does not admit a unique solution in general. Probabilistic methods are therefore a natural fit (Kaipio & Somersalo, 2005). Unlike existing blind inverse approaches that sample in data space (Chung et al., 2022) or are specialized to particular transformation families (Zitova & Flusser, 2003; Beg et al., 2005; Hartley et al., 2013), we tackle the problem for general Lie group actions, with all inference and sampling performed on the group.

We model inverse transformations using energy-based models which quantify the likelihood of different transformations of the data. This formulation allows us to leverage pretrained energy-based models or energies designed using domain expertise. Sampling from energy-based models can however be slow and challenging, especially when the energy landscape is rugged and multimodal. To address this, we introduce a new sampling method on Lie groups based on diffusion (Song et al., 2020), which follows scores along a noise schedule. We show that the sampling can be performed on the Lie group by estimating scores in the Lie algebra (Fig. 1), extending trivialization approaches for handling curved manifolds (Lezcano Casado, 2019; Zhu et al., 2025). Our method covers a very general class of groups without assuming compactness, a bi-invariant metric, or linear actions.

As a key application, we focus on the case where the transformed data is the input to a neural network. It is known that pretrained networks can exhibit a marked lack of robustness to geometric transformations (Hendrycks & Dietterich, 2019; Ollikka et al., 2024). Equivariant neural networks

$$p(g \mid \tilde{\mathbf{x}})$$
Posterior over transformations

Figure 1: Graphical model describing our problem and method (with observed variables in gray and unobserved variables in white). $\mathcal{X}$ denotes the data space and $G$ a group of transformations, here the group of image homographies $\mathrm{PGL}(3, \mathbb{R})$. We are provided a data sample $\tilde{\mathbf{x}}$ that is generated by transforming an unknown in-distribution sample $\mathbf{x}$ with an unknown transformation $g$. We wish to sample from the posterior over transformations. Inspired by diffusion models, we construct a fast sampler that reverts a diffusion process on the Lie group starting from a random group element. The scores of the stochastic differential equation (SDE) are computed in the Lie algebra $\mathfrak{g}$.

address this for some groups and modalities (Cohen & Welling, 2016; Ravanbakhsh et al., 2017; Kondor & Trivedi, 2018; Finzi et al., 2021), but require highly specialized architectures that can be difficult to scale. Instead of using equivariant models, we wish to improve the robustness of generic, pretrained networks. The ability to invert data transformation ensures that we can bring samples back in-distribution and that the model does not perform inference on out-of-distribution transformed data. Prior methods based on deterministic canonicalization (Jaderberg et al., 2015; Esteves et al., 2017; Kaba et al., 2023) pursue this idea but either rely on training additional equivariant networks (Mondal et al., 2023) or use optimization methods (Schmidt & Stober, 2024; Shumaylov et al., 2024; Singhal et al., 2025) which can be brittle and do not readily extend to general Lie groups.

Our method allows to make any pretrained model equivariant to Lie groups at test-time with access to an energy-based model. The energy can be approximated via the pretrained model itself (Grathwohl et al., 2019; Singhal et al., 2025), which results in a training-free pipeline. This can be interpreted as a form of inference-time scaling for equivariance. The inference-time cost comes from sampling in-distribution transformations, which is made efficient and fast using our novel diffusion sampler.

Our contributions are as follows:

1. We introduce Transformation-Inverting Energy Diffusion (TIED), a novel diffusion sampler for inverting data transformations. It is guaranteed to stay on manifold and handles general Lie groups and nonlinear actions, only requiring knowledge of Lie algebra.

2. We prove that inversion of data transformations can be used for test-time equivariance of pretrained models only requiring an approximate energy model.

3. We show experimentally that our method can invert transformations on challenging groups such as image homographies and Lie point symmetries for partial differential equations (PDEs). On these problems, we demonstrate that our method consistently outperforms baselines in improving the performance of pretrained networks.

## 2 BACKGROUND AND RELATED WORKS

### 2.1 INVERSE PROBLEMS ON GROUPS

Recovering unknown transformations has a long history in problems that are naturally formulated on a group. In alignment and rotation synchronization, the aim is to recover poses on orthogonal or Euclidean groups from pairs of relative measurements (Singer, 2011; Hartley et al., 2013). Related are also image registration problems in which one seeks to recover the affine or homography transformation between pairs of images (Zitova & Flusser, 2003; Beg et al., 2005; Lorenzi & Pennec, 2013). By contrast to our setting, in these problems we are often provided pairs of samples, as opposed to a single one. In addition, these methods typically rely on group-specific knowledge. A recent line of

work proposes to use generative models to recover inverse samples (Venkatakrishnan et al., 2013; Kadkhodaie & Simoncelli, 2020; Song et al., 2020; Chung et al., 2022). However, since they operate in data space instead of the group, they require large diffusion models and for non-linear actions do not recover transformations that lie on the manifold.

## 2.2 Test-time robustness and canonicalization

Canonicalization methods improve robustness of neural networks by mapping inputs to a reference pose before prediction (Jaderberg et al., 2015; Esteves et al., 2017; Kaba et al., 2023; Mondal et al., 2023). These methods offer an alternative paradigm to equivariant neural networks (Cohen & Welling, 2016; Bronstein et al., 2021) and data augmentation (Shorten & Khoshgoftaar, 2019; Cubuk et al., 2019) that does not require specialized architectures or more expensive training procedures. Canonicalization yields equivariance for any predictor if the canonicalization function is itself equivariant. Previous works have proposed probabilistic variants (Kim et al., 2023; Dym et al., 2024; Cornish, 2024; Lawrence et al., 2025), but they rely on equivariant networks and are therefore limited in applications. Most closely related to our method are optimization-based canonicalization methods (Kaba et al., 2023; Schmidt & Stober, 2024; Shumaylov et al., 2024; Singhal et al., 2025), which obtain an equivariant canonicalizer without requiring equivariant primitives. They are, however, purely deterministic, often specialize in specific groups and necessitate optimization of highly rugged energy functions. Test-time data augmentation methods are an alternative (Krizhevsky et al., 2012; Szegedy et al., 2015; Wang et al., 2019; Shanmugam et al., 2020; Kim et al., 2020), and improve prediction through an ensembling effect (Hansen & Salamon, 2002). They, however, lack equivariance guarantees and use heuristic augmentation distributions. Group averaging (Yarotsky, 2022) and its frame-based variant (Puny et al., 2021) guarantee equivariance, yet are often intractable for Lie groups and may average over out-of-distribution transformations.

## 2.3 Diffusion sampling

Diffusion models (Sohl-Dickstein et al., 2015; Ho et al., 2020; Song et al., 2020) are a class of generative models that produce samples by first sampling from a simple base distribution $p_1(\mathbf{x})$ and integrating along the stochastic differential equation (SDE)

$$\mathrm{d}\mathbf{x}_t = -\gamma(t)^2 \nabla_{\mathbf{x}_t} \log p_t(\mathbf{x}_t) \, \mathrm{d}t + \gamma(t) \, \mathrm{d}\bar{\mathbf{w}}_t, \tag{1}$$

where $t \in [0, 1]$, $\gamma(t)$ is a scalar diffusion coefficient, $\bar{\mathbf{w}}_t$ is Brownian motion run backwards in time. $p_t(\mathbf{x}_t)$ is a marginal 'noisy' distribution under the forward process

$$\mathrm{d}\mathbf{x}_t = \gamma(t) \, \mathrm{d}\mathbf{w}_t, \tag{2}$$

starting from the target distribution $p_0(\mathbf{x}) \equiv p(\mathbf{x})$. We consider here the variance-exploding (VE) SDE. The scores $\nabla_{\mathbf{x}_t} \log p_t(\mathbf{x}_t)$ can be estimated from data (Vincent, 2011). A key advantage of diffusion sampling is that the scores of the noisy densities are smoother than those of the original density and non-vanishing in regions of low energy (Song & Ermon, 2019). The success of diffusion models has inspired a variety of methods for sampling from Boltzmann distributions using reverse SDEs; see e.g., Richter & Berner (2023); Vargas et al. (2023); Akhound-Sadegh et al. (2024); De Bortoli et al. (2024). These methods enable the estimation of scores along the probability path, assuming access to the energy and its gradients, rather than samples. We discuss later extensions to Lie groups, which are relevant to our problem. Diffusion models over transformations have also been proposed (Bansal et al., 2023), but in contrast to us, use deterministic non-invertible transformations.

# 3 Probabilistic Transformation Inversion

## 3.1 Preliminaries

**Lie groups** We consider a set of transformations $G$ forming a connected Lie group. While a Lie group is a curved manifold, a lot of its properties derive from its tangent space at the identity, the Lie algebra, denoted by $\mathfrak{g} \equiv T_e G$. We denote by $\exp : \mathfrak{g} \to G$ the exponential map and by $\mu$ the (unnormalized) left-invariant Haar measure on $G$. The left multiplication map $L_g : G \to G$ is defined as $L_g : h \mapsto gh$, and its tangent map $\mathrm{d}(L_g)_h : T_h G \to T_{gh} G$ is one-to-one. We highlight two cases:

$$\mathrm{d}(L_g)_e : \mathfrak{g} \to T_g G, \quad \mathrm{d}(L_{g^{-1}})_g : T_g G \to \mathfrak{g}. \tag{3}$$

These tangent maps are useful for working in the Lie algebra instead of the tangent spaces of arbitrary group elements, a technique called *(left-)trivialization* in literature (Lezcano Casado, 2019; Kong & Tao, 2024). A particular example we frequently use is the trivialized gradient operator $\nabla_{\mathfrak{g}} \equiv \mathrm{d}(L_{g^{-1}})_g \nabla$, which takes a function on $G$ and evaluates its gradient in $\mathfrak{g}$. Notably, it can be computed with standard automatic differentiation tools without the need to explicitly handle tangent maps. We refer the reader to Appendix A for additional background on Lie groups and trivialization.

**Group actions** We consider data samples $\mathbf{x} \in \mathcal{X}$ and outputs $\mathbf{y} \in \mathcal{Y}$, where $\mathcal{X}, \mathcal{Y} \subseteq \mathbb{R}^d$. The data density is denoted $p_{\mathbf{x}}(\mathbf{x})$. We assume a diffeomorphic but not necessarily linear action of $g \in G$ on data samples $\phi_g(\mathbf{x})$ also denoted by $g \cdot \mathbf{x}$. The Jacobian of the group action $\phi_g$ evaluated at $\mathbf{x}$ is denoted by $J_g(\mathbf{x})$. For linear actions, $J_g$ is a group representation independent of $\mathbf{x}$. The orbit of $\mathbf{x}$ is the set of samples that can be obtained through transformations and is denoted by $G \cdot \mathbf{x}$. We denote by $\mu_{G \cdot \mathbf{x}}$ the pushforward of the Haar measure using $\phi_g(\mathbf{x})$. A function $f : \mathcal{X} \to \mathcal{Y}$ is equivariant if $f(g \cdot \mathbf{x}) = g \cdot f(\mathbf{x})$ for all $g \in G$, $\mathbf{x} \in \mathcal{X}$ and invariant if $f(g \cdot \mathbf{x}) = f(\mathbf{x})$. Similarly, a conditional density is equivariant if $p(g \cdot \mathbf{y} \mid g \cdot \mathbf{x}) = p(\mathbf{y} \mid \mathbf{x}) |\det J_g(\mathbf{y})|^{-1}$ for all $g \in G$, $\mathbf{x} \in \mathcal{X}$, $\mathbf{y} \in \mathcal{Y}$.

## 3.2 PROBLEM STATEMENT

We now formally introduce the problem of transformation inversion. We assume that we are given an out-of-distribution sample $\tilde{\mathbf{x}} = g \cdot \mathbf{x}$, generated by applying an unknown transformation $g$ on an in-distribution sample $\mathbf{x}$. Our aim is to solve the blind inverse problem and recover the pair $(g, \mathbf{x})$. Since this problem does not admit a unique solution, a probabilistic approach is a natural choice. Assuming a uniform prior over the unknown transformation, a classical solution to inverse problems (e.g., Stuart, 2010) is to consider the Bayesian posterior

$$p(\mathbf{x} \mid \tilde{\mathbf{x}}) = \frac{p_{\mathbf{x}}(\mathbf{x}) \, p(\tilde{\mathbf{x}} \mid \mathbf{x})}{p(\tilde{\mathbf{x}})} \propto p_{\mathbf{x}}(\mathbf{x}) \, \mathbb{1}_{G \cdot \mathbf{x}}(\tilde{\mathbf{x}}), \tag{4}$$

where the density is with respect to $\mu_{G \cdot \mathbf{x}}$ and $\mathbb{1}_{G \cdot \mathbf{x}}$ is the indicator function over the orbit of $\mathbf{x}$.

Rather than modeling the posterior in data space, we can alternatively model the posterior $p(g \mid \tilde{\mathbf{x}})$ directly on the group. This has the advantage that the group is (often significantly) lower dimensional than the data space, and that this allow to recover the inverse transformation $g$. We can show that the posterior is given by the following.

**Proposition 3.1** (Transformation inversion posterior). *Let $G$ act freely on $\mathcal{X}$. Then,*

1. *The posterior distribution of $g$ has density $p(g \mid \tilde{\mathbf{x}}) \propto p_{\mathbf{x}}(g^{-1} \cdot \tilde{\mathbf{x}}) |\det J_{g^{-1}}(\tilde{\mathbf{x}})|$.*

2. *The random variable $\mathbf{x}' = h^{-1} \cdot \tilde{\mathbf{x}}$, with $h \sim p(g \mid \tilde{\mathbf{x}})$ has density $p(\mathbf{x}' \mid \tilde{\mathbf{x}}) \propto p_{\mathbf{x}}(\mathbf{x}') \, \mathbb{1}_{G \cdot \mathbf{x}}(\tilde{\mathbf{x}})$ with respect to $\mu_{G \cdot \mathbf{x}}$.*

All the proofs appear in Appendix B. This result shows that when a prior model of data distribution $p_{\mathbf{x}}$ is available, the posterior over transformations can be directly recovered. Additionally, sampling $h \sim p(g \mid \tilde{\mathbf{x}})$ and canonicalizing the out-of-distribution sample via $h^{-1} \cdot \tilde{\mathbf{x}}$ yields samples from the prior data distribution, which conforms to intuition.

Writing the posterior distribution over $g$ as a Boltzmann density, we have

$$p(g \mid \tilde{\mathbf{x}}) = \frac{e^{-E_{\mathbf{x}}(g^{-1} \cdot \tilde{\mathbf{x}})} |\det J_{g^{-1}}(\tilde{\mathbf{x}})|}{Z(\tilde{\mathbf{x}})}, \quad Z(\tilde{\mathbf{x}}) = \int_G e^{-E_{\mathbf{x}}(g^{-1} \cdot \tilde{\mathbf{x}})} |\det J_{g^{-1}}(\tilde{\mathbf{x}})| \mathrm{d}\mu(g), \tag{5}$$

where $E_{\mathbf{x}}(\mathbf{x}) = -\log p_{\mathbf{x}}(\mathbf{x})$ is the energy associated with the data prior. Under mild conditions on the energy, the normalization constant $Z(\tilde{\mathbf{x}})$ is finite even for non-compact groups and the density is well-defined. Note that any function of $\tilde{\mathbf{x}}$ can be added to the energy without changing the density, since it will cancel in the normalization. Therefore, the energy does not need to be accurate across orbits, but only within each orbit. This is significantly simpler than modeling the data space and is closer to a generative model over transformations (Allingham et al., 2024).

## 3.3 APPLICATION TO TEST-TIME EQUIVARIANCE

We now consider an application of our framework in improving the robustness of a pretrained neural network $f_\theta : \mathcal{X} \to \mathcal{Y}$. A common failure mode of neural networks, including those trained on large

datasets and with data augmentation, is their poor generalization to transformed samples (Hendrycks & Dietterich, 2019). By investing a small amount of compute at test-time to sample an inverse transformation, we aim to improve the pretrained model.

Following the canonicalization method (Kaba et al., 2023), we define the test-time predictor $\tilde{f}_\theta$ as

$$\tilde{f}_\theta(\tilde{\mathbf{x}}) = g \cdot f\left(g^{-1} \cdot \tilde{\mathbf{x}}\right), \quad g \sim p\left(g \mid \tilde{\mathbf{x}}\right), \tag{6}$$

where input data to the pretrained model $g^{-1} \cdot \tilde{\mathbf{x}}$ is guaranteed to lie in distribution by Proposition 3.1. Results by Bloem-Reddy & Teh (2020); Lawrence et al. (2025) show that this randomized predictor is equivariant if $p\left(g \mid \tilde{\mathbf{x}}\right)$ is an equivariant conditional distribution. The natural question is therefore, does the transformation inversion posterior satisfy conditional equivariance? This is indeed the case, which ensures the soundness of our approach.

**Proposition 3.2** (Equivariance of the posterior). *For any $E_{\mathbf{x}} : \mathcal{X} \to \mathbb{R}$, the posterior density $p\left(g \mid \tilde{\mathbf{x}}\right)$ is $G$-equivariant.*

The intuition for this is that if the sample $\tilde{\mathbf{x}}$ is transformed, then the posterior must change in an opposite way to ensure that $g^{-1} \cdot \tilde{\mathbf{x}}$ would still be distributed according to the data prior.

Following Kim et al. (2023), we can also sample an ensemble of diverse in-distribution samples and average predictions over them when $p\left(g \mid \tilde{\mathbf{x}}\right)$ is equivariant:

$$\tilde{f}(\tilde{\mathbf{x}}) = \mathbb{E}_{g \sim p(g|\tilde{\mathbf{x}})}\left[g \cdot f\left(g^{-1} \cdot \tilde{\mathbf{x}}\right)\right]. \tag{7}$$

It has been shown that ensembling over augmentations can further improve the performance of pretrained neural networks (Shanmugam et al., 2020). Our method provides a principled form for the augmentation distribution $p\left(g \mid \tilde{\mathbf{x}}\right)$, whereas most methods use distributions motivated by heuristics.

There is significant flexibility in the choice of the energy function $E_{\mathbf{x}}$. Following optimization-based canonicalization methods (Schmidt & Stober, 2024; Singhal et al., 2025), we consider defining the energy directly from the pretrained model $f_\theta$ using its prediction confidence if it is a classifier (Grathwohl et al., 2019). We also consider the evidence lower-bound (ELBO) approximation of the energy using variational autoencoders (VAEs) (Kingma & Welling, 2013; Shumaylov et al., 2024).

## 4 TRANSFORMATION-INVERTING ENERGY DIFFUSION

### 4.1 CHALLENGES IN SAMPLING FROM GENERAL LIE GROUPS

The next question is how to sample from the transformation-inversion distribution Eq. (5), assuming we have access to an energy-based model and can take its gradient. This is challenging for two reasons. First, the energy landscape can be highly rugged and multi-modal, especially when the energy is derived from a neural network (Montúfar et al., 2014). This leads to slow mixing for MCMC, even for gradient-based methods such as Langevin sampling (Roberts & Tweedie, 1996; Welling & Teh, 2011). Second, we must sample on the Lie group and respect manifold constraints, while the energy is defined in data space rather than directly on the group. The first issue motivates the use of diffusion sampling methods, which have shown significant promise in sampling efficiently from Boltzmann densities. Fig. 2 shows an example of the benefits of diffusion for our problem. However, as we will see, existing methods do not address the second issue for general Lie groups.

### 4.2 DIFFUSION SAMPLING WITH TRIVIALIZED TARGET SCORE

We now present Transformation-Inverting Energy Diffusion (TIED), and its associated diffusion sampler on Lie groups, overcoming the aforementioned challenges. To simplify the notations, we fix a data $\tilde{\mathbf{x}}$ and write the transformation inversion posterior defined in Section 3.2 as $p(g) \equiv p(g \mid \tilde{\mathbf{x}})$. We also consider the energy $E : G \to \mathbb{R}$ associated with the posterior $p(g)$:

$$E(g) \equiv E_{\mathbf{x}}(g^{-1} \cdot \tilde{\mathbf{x}}) - \log|\det J_{g^{-1}}(\tilde{\mathbf{x}})|. \tag{8}$$

To sample from the posterior, we adopt *diffusion sampling*. The idea is to construct a *forward* noising process $p_t(g_t)$ on the group that gradually transforms the posterior $p_0 \equiv p$ into a simple noise distribution $p_1$. We then run this process *backwards in time* to turn noise samples into samples from the posterior.

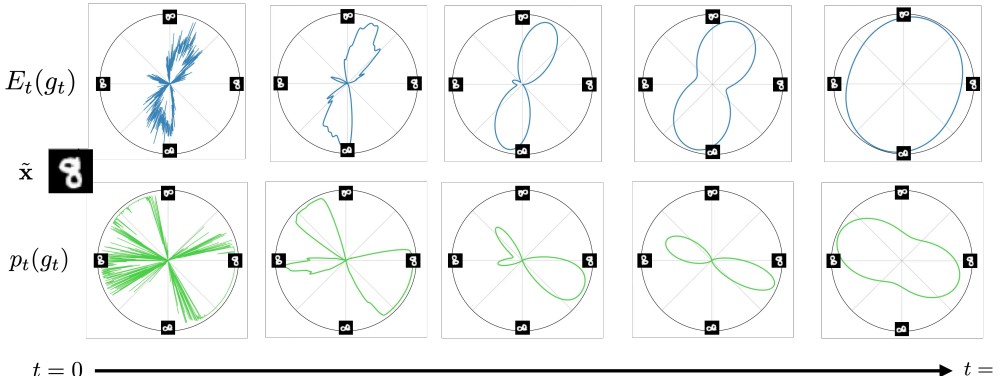

Figure 2: Energy (top) and density (bottom) along the forward process Eq. (9) for the group of rotations $G = \mathrm{SO}(2)$. The energy of the prior $E_0(g) \equiv E_{\mathbf{x}}(g^{-1} \cdot \tilde{\mathbf{x}})$ is defined using the LogSumExp of classifier logits from a ResNet18 trained on MNIST. The energy at small timesteps (top left) is low for likely orientations of the MNIST digit $\tilde{\mathbf{x}}$. Note the multimodal nature of the posterior $p_0(g)$ (bottom left), with modes centered around the most likely transformations $g = -20°$, $135°$, $180°$. Since the energy is obtained from a neural network, its landscape is highly rugged, resulting in exploding and vanishing scores. Going to the right, the densities along the forward process are plotted. The landscapes are increasingly smooth and address the ruggedness issue.

For the forward process, we propose to use a direct Lie-group analogue of the Euclidean variance-exploding SDE $\mathrm{d}\mathbf{x}_t = \gamma(t)\,\mathrm{d}\mathbf{w}_t$ in Eq. (2). The idea is to draw infinitesimal perturbations in the Lie algebra $\mathfrak{g}$ and then transport them to the current point on the group via right multiplication (i.e., trivialization):

$$\mathrm{d}g_t = \mathrm{d}(L_{g_t})_e\big[\gamma(t)\,\mathrm{d}\mathbf{w}_t^{\mathfrak{g}}\big], \quad g_0 \sim p, \tag{9}$$

where $\mathrm{d}\mathbf{w}_t^{\mathfrak{g}}$ is Brownian motion in the Lie algebra. Intuitively, we first draw a small random step in $\mathfrak{g}$, then use the tangent map $\mathrm{d}(L_g)_e : \mathfrak{g} \to T_g G$ to translate that step from the identity to the current location $g_t$ on the group. This plays the same role as adding Gaussian noise in the Euclidean setting.

A convenient property of this SDE is that its solution can be written in a simple *multiplicative* form: $g_t = g_0\,w_t$, where $w_t \sim k_t$ is a noise random variable on the group that is independent of $g_0$. This mirrors the Euclidean relationship $\mathbf{x}_t = \mathbf{x}_0 + \mathbf{w}_t$ for Gaussian noise $\mathbf{w}_t$, except that on a Lie group we combine the signal and noise via group multiplication rather than vector addition. While the distribution $k_t$ does not, in general, admit a closed-form expression because of the group geometry, we can still sample from it efficiently using the exponential map, which is sufficient for our purposes (see Appendix B.9).

For diffusion sampling, we reverse this construction: we start from noise samples $w_1 \sim k_1$ and run the SDE backwards in time to obtain samples from $p_0 \equiv p$. This is valid when the diffusion strengths $\gamma(t)$ are large enough that $k_1$ approximates the marginal $p_1$ of the forward process. We prove that this time-reversed dynamics is again a trivialized diffusion: it evolves on the group via Lie-algebra noise, but now with an additional drift term given by a *trivialized score* (score evaluated in the Lie algebra).

**Proposition 4.1** (Reverse trivialized SDE). *If each $p_t(g_t)$ is smooth and strictly positive with respect to the Haar measure, then the time-reversal of equation 9 is*

$$\mathrm{d}g_t = \mathrm{d}(L_{g_t})_e\big[-\gamma(t)^2\,\nabla_{\mathfrak{g}} \log p_t(g_t)\,\mathrm{d}t + \gamma(t)\,\mathrm{d}\bar{\mathbf{w}}_t^{\mathfrak{g}}\big], \tag{10}$$

*where $\bar{\mathbf{w}}_t^{\mathfrak{g}}$ is Brownian motion in $\mathfrak{g}$ run backwards in time, and $\nabla_{\mathfrak{g}} \log p_t(g_t)$ is the trivialized score, i.e., the score expressed as an element of the Lie algebra.*

Provided we can evaluate or estimate the trivialized score, diffusion sampling becomes straightforward: we discretize the reverse-time SDE and update group elements using the exponential map at each step. We present the resulting sampling scheme in Appendix B.9 and focus next on how to estimate the score.

A natural first idea for estimating the score is to learn it with a neural network via score matching (Huang et al., 2022; De Bortoli et al., 2022; Zhu et al., 2025). However, these methods typically

assume access to *clean* samples from $p_0 \equiv p$. In our setting, the posterior $p$ is only available through its associated energy $E \equiv -\log p$ rather than through samples. To address this problem, we introduce a new estimator of the trivialized score that uses the energy instead of requiring clean samples.

We build on the work of Akhound-Sadegh et al. (2024); De Bortoli et al. (2024), who use the *target score identity* to express scores of noisy distributions in terms of gradients of a clean energy. We present a new generalization of this identity to Lie groups that is compatible with trivialization:

**Proposition 4.2** (Trivialized target score identity). *For the forward SDE in equation 9, we have*

$$\nabla_{\mathfrak{g}} \log p_t(g_t) = \int_G \nabla_{\mathfrak{g}} \log p_0(g_0)\, p_{0|t}(g_0 \mid g_t)\, \mathrm{d}\mu(g_0) \tag{11}$$

*where the argument of $\log p_0$ is interpreted as $g_t b$ for $b \equiv g_t^{-1} g_0$, and $\nabla_{\mathfrak{g}}$ is taken with respect to $g_t$.*

In words, the identity says that the trivialized score at time $t$ can be written as an *average* of the initial score $\nabla_{\mathfrak{g}} \log p_0$ over all possible starting points $g_0$ that could have led to the current state $g_t$, weighted by the conditional density $p_{0|t}(g_0 \mid g_t)$. Since $\nabla_{\mathfrak{g}} \log p_0 = -\nabla_{\mathfrak{g}} E$ and we have access to $E$, this gives us a way to express the desired score entirely in terms of clean energy gradients.

A key feature of our result is that, unlike the Lie-group extension of the target score identity in De Bortoli et al. (2024, Section 3.2), it applies to *general* Lie groups and does not require a bi-invariant metric. The reason lies in the use of trivialization. By always expressing gradients as elements of the Lie algebra, we can meaningfully average energy gradients coming from different points on the group without explicitly transporting vectors between different tangent spaces. This removes the need for the right-invariance assumption used in De Bortoli et al. (2024, Appendix A.4) to handle such transports.

Based on the identity, we now derive a form of the trivialized score that is suitable for Monte Carlo estimation. The idea is to rewrite Eq. (11), currently weighted by the conditional density $p_{0|t}$, into an average weighted by the noise density $k_t$ which can be sampled easily. We achieve this by exploiting the multiplicative form $g_t = g_0 w$ for $w \sim k_t$, and performing a corresponding change of variables on the group. The derivation is in Appendix B.7, which is based on a nontrivial extension of the Euclidean case in Akhound-Sadegh et al. (2024, Section 3.1) to Lie groups.

**Proposition 4.3** (Monte Carlo score estimator). *For the forward SDE in equation 9, where $p_0$ is a Boltzmann density specified by a smooth energy $E$, we have*

$$\nabla_{\mathfrak{g}} \log p_t(g_t) = \nabla_{\mathfrak{g}} \log \int_G k_t(w) \exp\big(-E(g_t w^{-1}) - \log \lambda(w)\big) \mathrm{d}\mu(w) \tag{12}$$

*where $\lambda$ accounts for the change of the Haar measure under inversion, $\mathrm{d}\mu(w^{-1}) = \lambda(w)^{-1}\mathrm{d}\mu(w)$.*

This expression is well-suited for practical estimation. We can efficiently draw samples $w \sim k_t$, and the integral in Eq. (12) can then be approximated via Monte Carlo. We describe this procedure in detail and prove its consistency in Appendix B.8.

### 4.3 PRACTICAL IMPLEMENTATION

We now present a practical implementation of energy-based diffusion sampling with TIED (Algorithm 1). It runs time-discretized reverse SDE on Lie group using energy-based MC estimation of the trivialized score. Here, the number of samples $N$ is a hyperparameter which trades off computational cost and the quality of estimation. In our problem context of inverting transformations, we find that $N \lesssim 10$ usually yields a satisfactory performance and can be parallelized.

This implementation relies on a nice property of the score estimator that all its required components, including the volume correction $\lambda$ and the trivialized gradient $\nabla_{\mathfrak{g}}$, can be computed for general Lie groups using standard automatic differentiation tools. We explain these computations in detail in Appendix B.9.

---

**Algorithm 1:** Sampling with TIED

**Input:** Lie group $G$, energy $E$, noise schedule $\gamma$, step size $\Delta t$, Monte Carlo sample size $N$
**Output:** Sample $\hat{g}_0 \sim p \propto e^{-E(\cdot)}$

1   $M \leftarrow 1/\Delta t$
2   $\hat{g}_M \sim k_1$ using Eq. (84)
3   **for** $m \leftarrow M$ **to** $1$ **do**
4     $t \leftarrow m\Delta t$
5     **for** $i \leftarrow 1$ **to** $N$ **do**
6       $w^{(i)} \sim k_t$ using Eq. (84)
7       Compute $\log \lambda(w^{(i)})$ using Prop. B.13
8     **end**
9     $f(\cdot) \leftarrow \log \sum_i e^{-E\big((\cdot)\, w^{(i)-1}\big) - \log \lambda(w^{(i)})}$
10    $\hat{s}_m \leftarrow \nabla_{\mathfrak{g}} f(\hat{g}_m)$ using Prop. A.1
11    $\bar{z}_m \sim \mathcal{N}(0, \Delta t I)$
12    $\hat{g}_{m-1} \leftarrow \hat{g}_m \exp(\gamma(t)^2 \hat{s}_m \Delta t + \gamma(t)\bar{z}_m)$
13   **end**

---

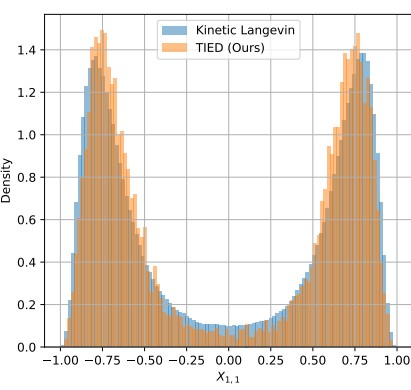 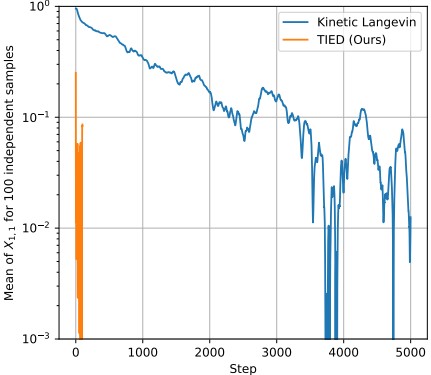

(a) Distributions of sampled $\mathbf{X}_{1,1}$.    (b) Mean of $\mathbf{X}_{1,1}$ over timesteps (lower the better).

Figure 3: Sampling on $\mathrm{SO}(10)$ under energy $E : \mathbf{X} \mapsto -10\mathbf{X}_{1,1}^2$ using a kinetic Langevin sampler (Kong & Tao, 2024) and TIED (Ours). The distribution of $\mathbf{X}_{1,1}$ induced by the energy has two symmetric modes around zero, and thus the mean of $\mathbf{X}_{1,1}$ approaches zero as the sampling converges.

## 5 EXPERIMENTS

We evaluate TIED on (i) a synthetic sampling problem on a high-dimensional Lie group; (ii) two image classification problems using a trained convolutional neural network under unknown affine and homography (perspective) transformations; and (iii) two partial differential equations (PDEs) solving problems using trained neural operators under unknown Lie point symmetry transformations. Supplementary experimental results are provided in Appendix C.

### 5.1 SYNTHETIC SAMPLING TASK ON $\mathrm{SO}(10)$

The first setting we consider is a synthetic energy-based sampling problem on a high-dimensional Lie group. We adopt the setup of Kong & Tao (2024) that considers the special orthogonal group $\mathrm{SO}(10)$ represented as a set of $10 \times 10$ matrices $\mathbf{X}$ with determinant 1. This group is chosen since its large intrinsic dimension $\dim \mathrm{SO}(10) = 45$ offers a challenge for sampling methods (by contrast to e.g. $\dim \mathrm{SO}(3) = 3$). The energy function we consider is $E : \mathbf{X} \mapsto -10\mathbf{X}_{1,1}^2$ where $\mathbf{X}_{1,1}$ is the value of the top-left matrix element. While the definition is simple, this energy induces a multimodal density $p \propto e^{-E(\cdot)}$ on the group, thereby offering a testbed aligning with our motivations in Section 4.1.

We compare TIED against the trivialized kinetic Langevin sampler proposed in Kong & Tao (2024), which only uses the energy gradient $\nabla_{\mathfrak{g}} E$. We run the Langevin sampler for 100,000 steps to ensure convergence. For TIED, we run 100 steps of diffusion, with sample size 100 for MC estimation of the score $\nabla_{\mathfrak{g}} \log p_t$. We provide the results in Figure 3, focusing on the top-left matrix element $\mathbf{X}_{1,1}$ for visualization. The results show that TIED samples from the correct multimodal distribution, and does so in a much fewer sampling steps, thanks to the diffusion formulation.

### 5.2 AFFINE AND HOMOGRAPHY INVARIANT IMAGE CLASSIFICATION

Next, we demonstrate TIED on image classification problems using a trained convolutional neural network under unknown affine and homography (perspective) transformations, closely following the experimental setup of Shumaylov et al. (2024). As our pretrained network $f_\theta$, we use a ResNet18 (He et al., 2016) trained to classify $40 \times 40$ padded MNIST images. To test its robustness and generalization, we consider two challenging transformation groups. The first is the group of affine transformations $\mathrm{Aff}(2, \mathbb{R})$, and the second is the group of homography (perspective) transformations, isomorphic to the projective general linear group $\mathrm{PGL}(3, \mathbb{R})$. Despite their widespread use in computer vision, these groups have high mathematical complexities: both are noncompact, non-Abelian, and lack a bi-invariant metric. We construct the respective test sets as 10,000-sized random subsets of affNIST and homNIST test images from MacDonald et al. (2022). In addition to classification accuracy of $f_\theta$, as a supplementary metric we measure the Fréchet inception distance (FID) (Heusel et al., 2017; Fatir, 2018) between the inverse-transformed test images the training images of $f_\theta$.

The baselines include general optimization and sampling methods on Lie groups, as well as specialized methods for inverting transformations for robust neural perception. For the former, we test trivialized kinetic Langevin (Kong & Tao, 2024) and Lie algebra canonicalization (LieLAC) (Shumaylov et al., 2024), which respectively do trivialized gradient-based sampling and optimization (Lezcano Casado, 2019). For the latter, we test inverse transformation search (ITS) (Schmidt & Stober, 2024), which performs an iterative search specifically designed for affine transforms, and FoCal (Singhal et al., 2025), which performs Bayesian optimization on transformation spaces (we use the Lie algebra).

Notably, all the baselines use an energy function (sometimes referred to as a data prior) to identify an inverse transformation, making it natural to compare them under the same choices of energy. For image classification, we experiment with two choices of energy functions, coarsely representative of probabilistic and predictive ones. For the probabilistic energy, we use ELBO of a VAE trained on clean MNIST images augmented with adversarial regularization, adopted from Shumaylov et al. (2024). For the predictive energy, we adopt the confidence-based energy measured by $-\texttt{logsumexp}$ of output logits (Grathwohl et al., 2019), similarly to Schmidt & Stober (2024); Singhal et al. (2025). For this, we use the same ResNet18 classifier $f_\theta$ for measuring the classification performance.

Table 1: MNIST classification test accuracy / FID.

| dataset | MNIST | | | |
|---|---|---|---|---|
| test transformations | none | | | |
| | | Acc | FID | |
| ResNet18 | | 99.35% | - | |
| test transformations | Aff$(2, \mathbb{R})$ | | PGL$(3, \mathbb{R})$ | |
| | Acc | FID | Acc | FID |
| affConv / homConv * | 95.08% | - | 95.71% | - |
| ResNet18 | 55.48% | - | 87.95% | - |
| Energy: VAE evidence lower bound (+ adv. reg.) | | | | |
| ResNet18 + ITS | 45.79% | 11.00 | n/a | n/a |
| ResNet18 + FoCal | 86.35% | 3.38 | 89.69% | 2.18 |
| ResNet18 + Kinetic Langevin | 74.55% | 7.15 | 93.72% | 0.77 |
| ResNet18 + LieLAC | 94.36% | 0.93 | 97.42% | **0.58** |
| ResNet18 + TIED (Ours) | **96.84%** | **0.71** | **97.45%** | **0.58** |
| Energy: Classifier logit confidence | | | | |
| ResNet18 + ITS | 69.03% | 9.88 | n/a | n/a |
| ResNet18 + FoCal | 66.73% | 10.88 | 86.24% | 3.92 |
| ResNet18 + Kinetic Langevin | 53.83% | 15.25 | 88.26% | 2.48 |
| ResNet18 + LieLAC | 73.58% | 6.97 | 85.91% | 2.63 |
| ResNet18 + TIED (Ours) | **85.21%** | **5.07** | **89.81%** | **1.83** |

\* from the original paper's table

The results are in Table 1. While the trained ResNet18 achieves high accuracy on clean images, affine or homography transformations substantially degrade it. Kinetic Langevin and LieLAC are based on (trivialized) gradients of the energy. Both restore the accuracy reasonably well, but relatively underperforms on affine using classifier energy. This is possibly due to the large magnitudes of transformations (as evidenced by the ResNet18 accuracies) and anomalies in the classifier's energy landscape affecting the gradients. In contrast, ITS and FoCal, which are iterative search methods, attain decent results on affine with classifier energy while often outperformed by gradient-based methods on others. We conjecture that their nonlocal iterative exploration helps handle the anomalies in the energy landscape. Finally, our method, TIED, combines precise local optimization via gradients with exploration by diffusion at high noise levels. As a result, it outperforms all baselines in all settings, in particular significantly improving the accuracy under affine transforms and classifier energy (55.48% → 82.51%). With a proper choice of the energy, TIED also outperforms specialized equivariant architectures affConv and homConv (MacDonald et al., 2022).

## 5.3 POINT SYMMETRY EQUIVARIANT PDE SOLVING

Next, we test TIED on a more challenging problem of solving partial differential equations (PDEs) with a trained neural operator under unknown Lie point symmetry transformations. Here, the model $f_\theta$ performs function-valued regression. At a high-level, the input to the model is a function evaluation $\mathbf{u}_0$ at a set of space and time points $(\mathbf{x}_0, t_0)$, which specifies an initial condition. The task is evaluating the function at another set of points $(\mathbf{x}_f, t_f)$, after $\mathbf{u}_0(\mathbf{x}_0, t_0)$ has evolved under the PDE of interest. More details can be found in Li et al. (2020); Lu et al. (2021).

Many PDEs possess a symmetry $G$ that transform a solution $\mathbf{u}(\mathbf{x}, t)$, to create other valid solutions (Olver, 1993; Brandstetter et al., 2022; Akhound-Sadegh et al., 2023). Since a neural operator $f_\theta$ is trained for a specific range of initial conditions $\mathbf{u}_0(\mathbf{x}_0, t_0)$ and prediction points $(\mathbf{x}_f, t_f)$, one expects it to fail outside the training domain. Akin to our problem setup, this can be overcome by finding a symmetry transformation $g$ which acts on input as $g^{-1} \cdot (\mathbf{u}_0(\mathbf{x}_0, t_0), \mathbf{x}_f, t_f)$ to push it into the training domain of $f_\theta$. Then, $f_\theta$ can make out-of-domain prediction as $g \cdot f_\theta(g^{-1} \cdot (\mathbf{u}_0(\mathbf{x}_0, t_0), \mathbf{x}_f, t_f))$. The main challenge is the complexity of Lie point symmetries: in many cases, they are noncompact and non-Abelian. This makes TIED an appealing candidate as it handles these cases.

We closely follow the setup of Shumaylov et al. (2024), using DeepONet neural operators (Lu et al., 2021) as the pretrained network $f_\theta$. To test the robustness and generalization, we use two PDEs with challenging symmetry groups. The first is 1D heat equation $u_t - \nu u_{xx} = 0$ with the symmetry

Table 2: PDE solving relative test L2 error.

| PDE | 1D Heat Eq. | 1D Heat Eq. + Data Aug. | 1D Burgers Eq. |
|---|---|---|---|
| Test transformations | none | none | none |
| DeepONet | 0.011 | 0.031 | 0.017 |
| Test transformations | $\mathrm{SL}(2,\mathbb{R}) \ltimes \mathrm{H}(1,\mathbb{R})$ | $\mathrm{SL}(2,\mathbb{R}) \ltimes \mathrm{H}(1,\mathbb{R})$ | $\mathrm{SL}(2,\mathbb{R}) \ltimes (\mathbb{R}^2,+)$ |
| DeepONet | 0.690 | 0.081 | 0.867 |
| Energy: Distance to training domain | | | |
| DeepONet + FoCal | $3.044 \pm 2.133$ | $1.618 \pm 1.081$ | $0.215 \pm 0.031$ |
| DeepONet + Kinetic Langevin | $0.587 \pm 0.049$ | $0.170 \pm 0.151$ | $0.745 \pm 0.035$ |
| DeepONet + LieLAC | $0.078 \pm 0.014$ | $0.088 \pm 0.013$ | $0.190 \pm 0.015$ |
| DeepONet + TIED (Ours) | $\mathbf{0.042 \pm 0.002}$ | $\mathbf{0.052 \pm 0.001}$ | $\mathbf{0.167 \pm 0.028}$ |

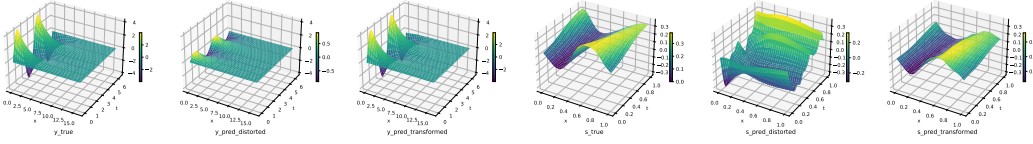

(a) 1D heat equation.          (b) 1D Burgers equation.

Figure 4: For each PDE, we show an out-of-domain case for DeepONet $f_\theta$. From left: true solution, $f_\theta$ prediction, $f_\theta$ prediction under test-time equivariance via TIED (Ours). **Zoom-in** for a better view.

group $\mathrm{SL}(2,\mathbb{R}) \ltimes \mathrm{H}(1,\mathbb{R})$, the semidirect product of special linear group and rank-one polarized Heisenberg group. The second is 1D Burgers' equation $u_t + uu_x - \nu u_x x = 0$ with the symmetry group $\mathrm{SL}(2,\mathbb{R}) \ltimes (\mathbb{R}^2,+)$. The test sets are constructed by applying these transformations to send $\mathbf{u}_0$ out of the training domain of $f_\theta$. For the choice of energy, we follow Shumaylov et al. (2024) and use a distance measure between the input $(\mathbf{u}_0(\mathbf{x}_0, t_0), \mathbf{x}_f, t_f)$ and the training domain of $f_\theta$.

The results are in Table 2 and Figure 4. While all DeepONets excel in their training domain, their performances degrade when presented with out-of-domain transformation, with data augmentation reducing the degradation but not perfectly. While FoCal performs poorly for heat equations, we find that it has a strong performance on Burgers equation. We conjecture that this is partially due to the Euclidean component $(\mathbb{R}^2, +)$ in the semidirect product, which may have created a more amenable environment for Bayesian optimization. While both based on gradients, LieLAC outperforms kinetic Langevin and attains good performances. This could be due to the Brownian motion term slowing down convergence. Our method TIED can leverage smoothened energy landscape at high noise levels effectively to speed up convergence, and achieves the best performance in all cases.

## 6 CONCLUSION

In this work, we studied the problem of inverting unknown data transformations from general Lie groups in a setting where we have access to a *data-space prior* specified by an energy function. Given such an energy in data space and a fixed pretrained model, our goal is to *transform* a test-time input so that it becomes well aligned with the model's training distribution and thereby achieves strong performance. This setting is increasingly relevant in the era of large foundation models, where robustness to out-of-distribution transformations remains a practical concern.

To address this, we proposed TIED, a purely test-time method that requires no training. TIED operates by reverting a diffusion process over transformations in the Lie algebra. Our approach naturally handles the rugged and multimodal energy landscapes that arise in practice, applies to a broad family of (possibly noncompact, non-Abelian) groups and nonlinear group actions, and remains computationally attractive by working in the low-dimensional space of transformations rather than the high-dimensional data space. We showed that sampling from the transformation posterior and canonicalizing the inputs at test time can reliably bring data back in-distribution and significantly improve the robustness of pretrained models without any retraining.

Multiple avenues remain for future work. One limitation of our method is that the score estimator requires evaluating the energy and its gradients over multiple MC samples. We could also consider accelerating the sampling, for example, using consistency distillation methods (Song et al., 2023). Applications to other inverse problems, such as image registration, would also prove interesting. Another potential extension is to problems where the group action is not known a priori and has to be learned (Koyama et al., 2023; Yang et al., 2023; Mitchel et al., 2024). Finally, a potentially exciting application of our method is to transformations that do not form a group. In practice, many relevant applications require dealing with transformations that are not invertible or inherently noisy.

## ETHICS STATEMENT

We have carefully reviewed the ICLR Code of Ethics and confirm that this work adheres to its principles. This work introduces a probabilistic framework and algorithm for inverting unknown data transformations from general Lie groups, aiming to improve the generalization of pretrained deep neural networks. Our contributions are theoretical and methodological, and we performed evaluations on publicly available benchmarks either in terms of artifacts or details of simulation. We do not anticipate direct ethical risks associated with this approach.

## REPRODUCIBILITY STATEMENT

We are committed to the reproducibility of our work. The appendix provides comprehensive details of the algorithm, including the numerical calculation of the proposed score estimator. We will release the source code upon acceptance to enable the replication of our results.

## USE OF LLMs

We used LLMs to find relevant theory, check grammar and improve the presentations of the sentences and materials in this paper. All final contents are carefully verified by the authors.

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

# A  BACKGROUND

We provide an overview of the mathematical background, and refer the readers to Lee (2012); Tu (2010) for more details.

**Lie group**  A Lie group $G$ is a group that is also a smooth manifold, such that multiplications of elements and taking inverses are smooth. In deep learning, it is useful as an abstraction of a class of continuous data transformations such as perspective transformations of an image. For any $g \in G$, we denote by $L_g : G \to G$ the left multiplication map $h \mapsto gh$ and by $R_g : G \to G$ the right multiplication map $h \mapsto hg$. We denote by $\mu$ the left-invariant (unnormalized) Haar measure on $G$.

**Tangent spaces**  As a manifold, a Lie group $G$ has a tangent space $T_g G$ at each point $g$, intuitively containing all possible directions of infinitesimal updates at $g$. We can map between tangent spaces using the notion of differentials. For any smooth function $f : G \to G$ and any $g \in G$, we denote by $\mathrm{d}f_g : T_g G \to T_{f(g)} G$ the differential of $f$ evaluated at $g$. For left multiplications $L_g$, the differentials $\mathrm{d}(L_g)_h : T_h G \to T_{gh} G$ are bijective linear maps between tangent spaces called the tangent maps. The tangent maps satisfy $\mathrm{d}(L_g)_{hk} \circ \mathrm{d}(L_h)_k = \mathrm{d}(L_{gh})_k$ and $\mathrm{d}(L_e)_g = \mathrm{id}_{T_g G}$ for all $g, h, k \in G$.

**Lie algebra**  While a Lie group is a curved manifold, a lot of its properties derive from its tangent space at the identity, the Lie algebra, defined by $\mathfrak{g} \equiv T_e G$. For an $n$-dimensional Lie group, the Lie algebra $\mathfrak{g}$ is an $n$-dimensional vector space equipped with a binary operation $[\cdot, \cdot] : \mathfrak{g} \times \mathfrak{g} \to \mathfrak{g}$ called the Lie bracket. We can always construct an inner product $\langle \cdot, \cdot \rangle_{\mathfrak{g}}$ on $\mathfrak{g}$ by choosing any basis $\{\mathbf{e}_1, ..., \mathbf{e}_n\}$ and defining $\langle \sum_i u_i \mathbf{e}_i, \sum_i v_i \mathbf{e}_i \rangle_{\mathfrak{g}} \equiv \sum_i u_i v_i$. This is the unique inner product on $\mathfrak{g}$ for which the chosen basis is orthonormal. The exponential map $\exp : \mathfrak{g} \to G$ is a smooth map that specifies group structure from the Lie algebra. In general, this specification is local around the identity $e = \exp(\mathbf{0})$ as $\exp$ can be non-surjective for non-compact $G$. Yet, for connected Lie groups, any element can be reached as a finite product of these local elements (Olver, 1993, Proposition 1.24).

**Gradients**  We can define the notion of gradients on a Lie group by equipping it with a choice of metric. For a finite-dimensional Lie group, one possible choice is a metric $\langle \cdot, \cdot \rangle$ that is left-invariant:

$$\langle \mathrm{d}(L_h)_g \mathbf{u}, \mathrm{d}(L_h)_g \mathbf{v} \rangle_{hg} = \langle \mathbf{u}, \mathbf{v} \rangle_g. \tag{13}$$

Such a metric can be always constructed by extending any inner product $\langle \cdot, \cdot \rangle_{\mathfrak{g}}$ on the Lie algebra:

$$\langle \mathbf{u}, \mathbf{v} \rangle_g \equiv \langle \mathrm{d}(L_{g^{-1}})_g \mathbf{u}, \mathrm{d}(L_{g^{-1}})_g \mathbf{v} \rangle_{\mathfrak{g}}. \tag{14}$$

Then, for any smooth function $f : G \to \mathbb{R}$, the gradient $\nabla f(g)$ at each point $g \in G$ is given as the unique element of the tangent space $T_g G$ that satisfies the following for all $\mathbf{v} \in T_g G$:

$$\langle \nabla f(g), \mathbf{v} \rangle_g = \mathrm{d}f_g(\mathbf{v}), \tag{15}$$

where $\mathrm{d}f_g : T_g G \to \mathbb{R}$ is the differential of $f$ evaluated at $g$ (Lee, 2012, Section 13).

**Trivialization**  In deep learning context, the curved geometry of Lie groups can be challenging to deal with. We can narrow this down to two specific difficulties in our problem setup.

1. In energy-based diffusion, one usually employs an expectation form of the score function that averages energy gradients evaluated across the state space (De Bortoli et al., 2024; Akhound-Sadegh et al., 2024). However, on a Lie group, gradients of a function at different points live in different tangent spaces and hence are not directly compatible, e.g., cannot be added or averaged.

2. In general, during sampling on a Lie group, the update directions live in tangent spaces that change over time simultaneously as a sample is updated, requiring a careful handling.

We can sidestep both difficulties by always working in the Lie algebra instead of arbitrary tangent spaces. This technique is called (left-)trivialization (Lezcano Casado, 2019; Tao & Ohsawa, 2020; Kong & Tao, 2024; Zhu et al., 2025). We use two related tools that together address the difficulties.

Our first tool is **trivialized gradient** that can be understood as gradient evaluated in the Lie algebra. For any smooth function $f : G \to \mathbb{R}$, we define the trivialized gradient $\nabla_{\mathfrak{g}} f(g) \in \mathfrak{g}$ at each point $g \in G$ by mapping the standard gradient $\nabla f(g) \in T_g G$ through the tangent map $\mathrm{d}(L_{g^{-1}})_g : T_g G \to \mathfrak{g}$:

$$\nabla_{\mathfrak{g}} f(g) \equiv \mathrm{d}(L_{g^{-1}})_g \nabla f(g). \tag{16}$$

Since trivialized gradients always live in the same vector space $\mathfrak{g}$, we can add or average them without restrictions. This underlies our novel score identity on general Lie groups.

Another nice property of trivialized gradients is that, despite their definition involving a tangent map, they can be computed directly using automatic differentiation without any explicit handling of tangent maps. This knowledge is not new (Kobilarov, 2014a;b), but rarely used in deep learning context.

**Proposition A.1** (Trivialized gradient). *Let $G$ be a finite-dimensional Lie group. Under any choice of a left-invariant metric, for any smooth function $f : G \to \mathbb{R}$, the following holds for all $g \in G$:*

$$\nabla_{\mathfrak{g}} f(g) = \nabla_{\mathbf{v}} f(g \exp(\mathbf{v}))|_{\mathbf{v}=\mathbf{0}} . \tag{17}$$

*Proof.* The proof is given in Appendix B.3. $\square$

Also, as a useful property, we can use the standard log-derivative trick on trivialized gradients:

**Proposition A.2** (Trivialized score). *Let $G$ be a finite-dimensional Lie group. Under any choice of a left-invariant metric, for any positive and smooth function $p : G \to \mathbb{R}$, it holds that $\nabla_{\mathfrak{g}} p = p \nabla_{\mathfrak{g}} \log p$.*

*Proof.* The proof is given in Appendix B.4. $\square$

Our second tool is **trivialized SDE** that specifies diffusion on Lie groups using components in the Lie algebra. Specifically, for an $n$-dimensional Lie group $G$ and any choice of an orthonormal basis of the Lie algebra $\mathfrak{g}$, we consider Itô SDEs on the group having the following form:

$$\mathrm{d}g_t = \mathrm{d}(L_{g_t})_e \left[ \phi(g_t, t) \, \mathrm{d}t + \gamma(t) \, \mathrm{d}\mathbf{w}_t^{\mathfrak{g}} \right] , \tag{18}$$

with $\mathrm{d}\mathbf{w}_t^{\mathfrak{g}}$ the Brownian motion on Lie algebra given as $\mathrm{d}\mathbf{w}_t^{\mathfrak{g}} = \sum_i \mathbf{e}_i \mathrm{d}w_t^i$ where $\mathrm{d}w_t^1, ..., \mathrm{d}w_t^n$ are independent standard Brownian motions on $\mathbb{R}$ and $\{\mathbf{e}_1, ..., \mathbf{e}_n\}$ is the chosen orthonormal basis of $\mathfrak{g}$. In addition, $\phi(\cdot, t) : G \to \mathfrak{g}$ is the drift coefficient, and $\gamma(t) \in \mathbb{R}$ is the diffusion coefficient.

The trivialized nature of the SDE motivates a natural Euler time-discretization scheme for sampling that eliminates the need to handle moving tangent spaces. This is based on the fact that, for any fixed $\mathbf{v} \in \mathfrak{g}$, the trivialized dynamics $\mathrm{d}g = \mathrm{d}(L_g)_e \mathbf{v} \, \mathrm{d}t$ has a closed-form solution $g(t) = g(0) \exp(t\mathbf{v})$ that can be computed using the exponential map (Lee, 2012, (8.13) and Proposition 20.8). Specifically, for step size $\Delta t > 0$, a Euler time-discretization can be written as follows, where $m = 0, ..., 1/\Delta t$:

$$\hat{g}_{m+1} = \hat{g}_m \exp \left( \phi(\hat{g}_m, m\Delta t) \, \Delta t + \gamma(m\Delta t) \, \mathbf{z}_m \right), \quad \mathbf{z}_m \sim \mathcal{N}(\mathbf{0}, \Delta t \mathbf{I}), \quad \hat{g}_0 \stackrel{d}{=} g_0. \tag{19}$$

## B PROOFS

### B.1 PROOF OF PROPOSITION 3.1

**Proposition 3.1** (Transformation inversion posterior). *Let $G$ act freely on $\mathcal{X}$. Then,*

1. *The posterior distribution of $g$ has density $p\left(g \mid \tilde{\mathbf{x}}\right) \propto p_{\mathbf{x}}\left(g^{-1} \cdot \tilde{\mathbf{x}}\right)|\det J_{g^{-1}}\left(\tilde{\mathbf{x}}\right)|$.*

2. *The random variable $\mathbf{x}' = h^{-1} \cdot \tilde{\mathbf{x}}$, with $h \sim p\left(g \mid \tilde{\mathbf{x}}\right)$ has density $p\left(\mathbf{x}' \mid \tilde{\mathbf{x}}\right) \propto p_{\mathbf{x}}\left(\mathbf{x}'\right) \mathbb{1}_{G \cdot \mathbf{x}}\left(\tilde{\mathbf{x}}\right)$ with respect to $\mu_{G \cdot \mathbf{x}}$.*

*Proof.* For the first statement, using independence of $\mathbf{x}$ and $g$ and the relation $\tilde{\mathbf{x}} = g \cdot \mathbf{x}$, we have

$$p\left(\mathbf{x}, \tilde{\mathbf{x}} \mid g\right) = p_{\mathbf{x}}\left(\mathbf{x}\right) p\left(\tilde{\mathbf{x}} \mid \mathbf{x}, g\right) \tag{20}$$

$$p\left(\mathbf{x}, \tilde{\mathbf{x}} \mid g\right) = p_{\mathbf{x}}\left(\mathbf{x}\right) \delta\left(\tilde{\mathbf{x}} - g \cdot \mathbf{x}\right) \tag{21}$$

Marginalizing with respect to $\mathbf{x}$ yields

$$p\left(\tilde{\mathbf{x}} \mid g\right) = \int p_{\mathbf{x}}\left(\mathbf{x}\right) \delta\left(\tilde{\mathbf{x}} - g \cdot \mathbf{x}\right) \mathrm{d}\mathbf{x} \tag{22}$$

Since the group action is diffeomorphic, we have by the change of variable formula

$$p\left(\tilde{\mathbf{x}} \mid g\right) = \int p_{\mathbf{x}}\left(\mathbf{x}\right) \delta\left(g^{-1} \cdot \tilde{\mathbf{x}} - \mathbf{x}\right) |\det J_{g^{-1}}\left(\tilde{\mathbf{x}}\right)| \mathrm{d}\mathbf{x} \tag{23}$$

$$p\left(\tilde{\mathbf{x}} \mid g\right) = p_{\mathbf{x}}\left(g^{-1} \cdot \tilde{\mathbf{x}}\right) |\det J_{g^{-1}}\left(\tilde{\mathbf{x}}\right)| \tag{24}$$

With a uniform prior on $g$, we obtain via Bayes rule

$$p\left(g \mid \tilde{\mathbf{x}}\right) \propto p_{\mathbf{x}}\left(g^{-1} \cdot \tilde{\mathbf{x}}\right) |\det J_{g^{-1}}\left(\tilde{\mathbf{x}}\right)| \tag{25}$$

For the second statement, for $\mathbf{x}' = g^{-1} \cdot \tilde{\mathbf{x}}$ and measurable $B \subseteq G \cdot \tilde{\mathbf{x}}$, we have

$$\mathbb{P}\left(\mathbf{x}' \in B \mid \tilde{\mathbf{x}}\right) = \int_{g \in G} p\left(g \mid \tilde{\mathbf{x}}\right) \mathbb{1}_B\left(g^{-1} \cdot \tilde{\mathbf{x}}\right) \mathrm{d}\mu(g) \tag{26}$$

$$\mathbb{P}\left(\mathbf{x}' \in B \mid \tilde{\mathbf{x}}\right) \propto \int_{g \in G} p_{\mathbf{x}}\left(g^{-1} \cdot \tilde{\mathbf{x}}\right) |\det J_{g^{-1}}\left(\tilde{\mathbf{x}}\right)| \mathbb{1}_B\left(g^{-1} \cdot \tilde{\mathbf{x}}\right) \mathrm{d}\mu(g) \tag{27}$$

Making the change of variable $\mathbf{x}' = g^{-1} \cdot \tilde{\mathbf{x}}$ and using the fact that the group action is diffeomorphic, we have by the change of variable formula

$$\mathbb{P}\left(\mathbf{x}' \in B \mid \tilde{\mathbf{x}}\right) \propto \int_{\mathbf{x}' \in G \cdot \tilde{\mathbf{x}}} p_{\mathbf{x}}\left(\mathbf{x}'\right) \mathbb{1}_B\left(\mathbf{x}'\right) \mathrm{d}\mu_{G \cdot \tilde{\mathbf{x}}}(\mathbf{x}') \tag{28}$$

Therefore we have the following density

$$p\left(\mathbf{x}' \mid \tilde{\mathbf{x}}\right) \propto p_{\mathbf{x}}\left(\mathbf{x}'\right) \mathbb{1}_{G \cdot \tilde{\mathbf{x}}}\left(\mathbf{x}'\right) \tag{29}$$

with respect to $\mu_{G \cdot \tilde{\mathbf{x}}}$. $\square$

### B.2 PROOF OF PROPOSITION 3.2

**Proposition 3.2** (Equivariance of the posterior). *For any $E_{\mathbf{x}} : \mathcal{X} \to \mathbb{R}$, the posterior density $p\left(g \mid \tilde{\mathbf{x}}\right)$ is $G$-equivariant.*

*Proof.* By left-invariance of the Haar measure, the density $p\left(g \mid \tilde{\mathbf{x}}\right)$ is equivariant if

$$p\left(h \cdot g \mid h \cdot \tilde{\mathbf{x}}\right) = p\left(g \mid \tilde{\mathbf{x}}\right), \quad \forall g, h \in G, \tilde{\mathbf{x}} \in \mathcal{X} \tag{30}$$

We can verify this explicitly for the posterior given by

$$p\left(g \mid \tilde{\mathbf{x}}\right) = \frac{e^{-E_{\mathbf{x}}\left(g^{-1} \cdot \tilde{\mathbf{x}}\right)} |\det J_{g^{-1}}\left(\tilde{\mathbf{x}}\right)|}{Z\left(\tilde{\mathbf{x}}\right)} \tag{31}$$

We have

$$p\left(h \cdot g \mid h \cdot \tilde{\mathbf{x}}\right) = \frac{e^{-E_{\mathbf{x}}\left(g^{-1} \cdot h^{-1} \cdot h \cdot \tilde{\mathbf{x}}\right)} |\det J_{(h \cdot g)^{-1}}\left(h \cdot \tilde{\mathbf{x}}\right)|}{Z\left(h \cdot \tilde{\mathbf{x}}\right)} \tag{32}$$

$$p\left(h \cdot g \mid h \cdot \tilde{\mathbf{x}}\right) = \frac{e^{-E_{\mathbf{x}}\left(g^{-1} \cdot \tilde{\mathbf{x}}\right)} |\det J_{(h \cdot g)^{-1}}\left(h \cdot \tilde{\mathbf{x}}\right)|}{Z\left(h \cdot \tilde{\mathbf{x}}\right)} \tag{33}$$

Using the chain rule for the Jacobian, we obtain

$$p\left(h \cdot g \mid h \cdot \tilde{\mathbf{x}}\right) = \frac{e^{-E_{\mathbf{x}}\left(g^{-1} \cdot \tilde{\mathbf{x}}\right)} |\det J_{g^{-1}}\left(\tilde{\mathbf{x}}\right)| |\det J_{h^{-1}}\left(h \cdot \tilde{\mathbf{x}}\right)|}{Z\left(h \cdot \tilde{\mathbf{x}}\right)} \tag{34}$$

We can absorb the factor $|\det J_{h^{-1}}\left(h \cdot \tilde{\mathbf{x}}\right)|$ into the normalization constant by using the chain rule for Jacobians, then a change of variable $g' = h^{-1} g$ followed by left-invariance of the Haar measure

$$\frac{|\det J_{h^{-1}}(h \cdot \tilde{\mathbf{x}})|}{Z(h \cdot \tilde{\mathbf{x}})} = \frac{|\det J_{h^{-1}}(h \cdot \tilde{\mathbf{x}})|}{\int_G e^{-E_{\mathbf{x}}(g^{-1}h \cdot \tilde{\mathbf{x}})} |\det J_{g^{-1}}(h \cdot \tilde{\mathbf{x}})| \mathrm{d}\mu(g)}$$

$$= \frac{|\det J_{h^{-1}}(h \cdot \tilde{\mathbf{x}})|}{\int_G e^{-E_{\mathbf{x}}(g^{-1}h \cdot \tilde{\mathbf{x}})} |\det J_{g^{-1}h}(\tilde{\mathbf{x}})| |\det J_{h^{-1}}(h \cdot \tilde{\mathbf{x}})| \mathrm{d}\mu(g)}$$

$$= \frac{1}{\int_G e^{-E_{\mathbf{x}}(g'^{-1} \cdot \tilde{\mathbf{x}})} |\det J_{g'^{-1}}(\tilde{\mathbf{x}})| \mathrm{d}\mu(hg')}$$

$$= \frac{1}{Z(\tilde{\mathbf{x}})} \tag{35}$$

This gives us $p\left(h \cdot g \mid h \cdot \tilde{\mathbf{x}}\right) = p\left(g \mid \tilde{\mathbf{x}}\right)$. $\square$

### B.3 PROOF OF PROPOSITION A.1

**Proposition A.1** (Trivialized gradient). *Let $G$ be a finite-dimensional Lie group. Under any choice of a left-invariant metric, for any smooth function $f : G \to \mathbb{R}$, the following holds for all $g \in G$:*

$$\nabla_{\mathfrak{g}} f(g) = \nabla_{\mathbf{v}} f(g \exp(\mathbf{v}))|_{\mathbf{v}=\mathbf{0}}. \tag{36}$$

*Proof.* Let $\mathbf{u} \in \mathfrak{g}$ be an arbitrary unit vector and let $\gamma(t) \equiv g \exp(t\mathbf{u})$ be an associated smooth curve on the group parameterized by $t \in \mathbb{R}$. Then from Lee (2012, Corollary 3.25) the following holds

$$\mathrm{d}f_g(\dot{\gamma}(0)) = \left.\frac{\mathrm{d}}{\mathrm{d}t} f(\gamma(t))\right|_{t=0} \tag{37}$$

Starting from the left-hand side, we get

$$\begin{aligned}
\mathrm{d}f_g(\dot{\gamma}(0)) &= \langle \nabla f(g), \dot{\gamma}(0) \rangle_g \\
&= \langle \nabla f(g), \mathrm{d}(L_g)_e \mathbf{u} \rangle_g \\
&= \langle \mathrm{d}(L_{g^{-1}})_g \nabla f(g), \mathrm{d}(L_{g^{-1}})_g \circ \mathrm{d}(L_g)_e \mathbf{u} \rangle_e \\
&= \langle \nabla_{\mathfrak{g}} f(g), \mathbf{u} \rangle_e
\end{aligned} \tag{38}$$

where we use $\dot{\gamma}(0) = \mathrm{d}(L_{\gamma(0)})_e \mathbf{u} = \mathrm{d}(L_g)_e \mathbf{u}$ (Lee, 2012, Sections 8, 9, 20) for the second equality.

Starting from the right-hand side, we get

$$\begin{aligned}
\left.\frac{\mathrm{d}}{\mathrm{d}t} f(\gamma(t))\right|_{t=0} &= \left.\frac{\mathrm{d}}{\mathrm{d}t} f(g \exp(\mathbf{0} + t\mathbf{u}))\right|_{t=0} \\
&= \nabla_{\mathbf{u}}(f \circ L_g \circ \exp)(\mathbf{0}) \\
&= \langle \nabla_{\mathbf{v}} f(g \exp(\mathbf{v}))|_{\mathbf{v}=\mathbf{0}}, \mathbf{u} \rangle_e
\end{aligned} \tag{39}$$

where, for the second equality, we use the definition of directional derivative of the composite function $f \circ L_g \circ \exp : \mathfrak{g} \to \mathbb{R}$ along the direction $\mathbf{u}$ evaluated at $\mathbf{0}$.

Therefore, for arbitrary unit vector $\mathbf{u} \in \mathfrak{g}$ it holds that

$$\langle \nabla_{\mathfrak{g}} f(g), \mathbf{u} \rangle_e = \langle \nabla_{\mathbf{v}} f(g \exp(\mathbf{v}))|_{\mathbf{v}=\mathbf{0}}, \mathbf{u} \rangle_e \tag{40}$$

and we have equation 36. $\qquad\square$

### B.4 PROOF OF PROPOSITION A.2

**Proposition A.2** (Trivialized score). *Let $G$ be a finite-dimensional Lie group. Under any choice of a left-invariant metric, for any positive and smooth function $p : G \to \mathbb{R}$, it holds that $\nabla_{\mathfrak{g}} p = p \nabla_{\mathfrak{g}} \log p$.*

*Proof.* For any smooth function $f : G \to \mathbb{R}$, we can turn its differential $\mathrm{d}f_g$ evaluated at $g$ into a linear functional on $\mathfrak{g}$ by pre-composing with $\mathrm{d}(L_{g^{-1}})_g^{-1}$. Concretely, for any $g \in G$ and $\mathbf{u} \in \mathfrak{g}$,

$$\langle \nabla_{\mathfrak{g}} f(g), \mathbf{u} \rangle_{\mathfrak{g}} = \mathrm{d}f_g(\mathrm{d}(L_{g^{-1}})_g^{-1} \mathbf{u}) = \left.\frac{\mathrm{d}}{\mathrm{d}t} f(g \exp(t\mathbf{u}))\right|_{t=0} \tag{41}$$

This is exactly the directional derivative of $f$ along the left-invariant vector field generated by $\mathbf{u}$.

We can apply the above equation to the left-trivialized score since we assumed the smoothness of each $p$. With $f = \log p$, we get

$$(\mathrm{d}(L_{g^{-1}})_g \nabla \log p(g))(\mathbf{u}) = \left.\frac{\mathrm{d}}{\mathrm{d}t} \log p(g \exp(t\mathbf{u}))\right|_{t=0} \tag{42}$$

By the chain rule on $\mathbb{R}$,

$$\frac{\mathrm{d}}{\mathrm{d}t} \log p(g \exp(t\mathbf{u})) = \frac{1}{p(g \exp(t\mathbf{u}))} \frac{\mathrm{d}}{\mathrm{d}t} p(g \exp(t\mathbf{u})) \tag{43}$$

Evaluating at $t = 0$ gives

$$\langle \nabla_{\mathfrak{g}} \log p(g), \mathbf{u} \rangle_{\mathfrak{g}} = \frac{1}{p(g)} \left.\frac{\mathrm{d}}{\mathrm{d}t} p(g \exp(t\mathbf{u}))\right|_{t=0} = \frac{\langle \nabla_{\mathfrak{g}} p(g), \mathbf{u} \rangle_{\mathfrak{g}}}{p(g)} \tag{44}$$

Since this holds for arbitrary $\mathbf{u} \in \mathfrak{g}$, we get $\nabla_{\mathfrak{g}} p(g) = p(g) \nabla_{\mathfrak{g}} \log p(g)$ for all $g \in G$. $\qquad\square$

## B.5 PROOF OF PROPOSITION 4.1

The proof is inspired by the generator-based approach of Zhu et al. (2025); Holderrieth et al. (2024).

For an $n$-dimensional Lie group $G$ with any choice of orthonormal basis $\{\mathbf{e}_1, ..., \mathbf{e}_n\}$ of Lie algebra $\mathfrak{g}$, we recall the SDE given in equation 18, with Brownian motion in the Lie algebra $\mathrm{d}\mathbf{w}_t^{\mathfrak{g}} = \sum_i \mathbf{e}_i \mathrm{d}w_t^i$, drift coefficient $\phi(\cdot, t) : G \to \mathfrak{g}$, and diffusion coefficient $\gamma(t) \in \mathbb{R}$:

$$\mathrm{d}g_t = \mathrm{d}(L_{g_t})_e \left[ \phi(g_t, t) \, \mathrm{d}t + \gamma(t) \, \mathrm{d}\mathbf{w}_t^{\mathfrak{g}} \right]. \tag{45}$$

We show some useful lemmas related to the SDE. We first derive its infinitesimal generator $\mathcal{L}_t$, which is a linear operator defined for every smooth function $f : G \to \mathbb{R}$ as the following for each $g \in G$:

$$(\mathcal{L}_t f)(g) \equiv \lim_{h \to 0^+} \frac{\mathbb{E}[f(g_{t+h}) - f(g_t)|g_t = g]}{h}. \tag{46}$$

**Lemma B.5.** *Assume the SDE in equation 18 has a smooth drift. Then its infinitesimal generator $\mathcal{L}_t$ satisfies the following for any smooth $f : G \to \mathbb{R}$, where $\Delta$ is the Laplace-Beltrami operator:*

$$\mathcal{L}_t f = \langle \nabla_{\mathfrak{g}} f, \phi(\cdot, t) \rangle_{\mathfrak{g}} + \frac{\gamma(t)^2}{2} \Delta f. \tag{47}$$

*Proof.* Let us denote $E_i \equiv \mathrm{d}(L_{[\cdot]})_e \mathbf{e}_i$ and rewrite equation 18 as follows

$$\mathrm{d}g_t = \Phi(g_t, t) \, \mathrm{d}t + \sum_{i=1}^n \Gamma_i(g_t, t) \, \mathrm{d}w_t^i, \quad \Phi(g, t) \equiv \mathrm{d}(L_g)_e \phi(g, t), \quad \Gamma_i(g, t) \equiv \gamma(t) E_i \tag{48}$$

Since $\Phi(\cdot, t), \Gamma_1(\cdot, t), ..., \Gamma_n(\cdot, t)$ are smooth vector fields on $G$, this expression makes it explicit that equation 18 is an Itô SDE taking $G$ as the state space. Then, by applying Lee & Chirikjian (2025, Theorem 2 and Proposition 1)[1] we get

$$\begin{aligned}
\mathcal{L}_t f &= \mathrm{d}f_{[\cdot]}(\Phi(\cdot, t)) + \frac{1}{2} \sum_{i=1}^n \mathrm{Hess}_f(\Gamma_i(\cdot, t), \Gamma_i(\cdot, t)) \\
&= \mathrm{d}f_{[\cdot]}(\Phi(\cdot, t)) + \frac{\gamma(t)^2}{2} \sum_{i=1}^n \mathrm{Hess}_f(E_i, E_i) \\
&= \mathrm{d}f_{[\cdot]}(\Phi(\cdot, t)) + \frac{\gamma(t)^2}{2} \Delta f \\
&= \langle \nabla f, \Phi(\cdot, t) \rangle + \frac{\gamma(t)^2}{2} \Delta f
\end{aligned} \tag{49}$$

where we use bilinearity of Hessian for the second equality and use equation 15 for the last equality. With left-invariance of the metric we obtain equation 47. $\square$

We now derive the adjoint operator $\mathcal{L}_t^*$ of the infinitesimal generator, which is defined through the following relationship for any smooth, compactly supported test function $f : G \to \mathbb{R}$ and density $\rho$ with respect to the Haar measure $\mu$ (Holderrieth et al., 2024, Appendix A.3):

$$\int_G (\mathcal{L}_t f) \, \rho \, \mathrm{d}\mu = \int_G f \, (\mathcal{L}_t^* \rho) \, \mathrm{d}\mu. \tag{50}$$

**Lemma B.6.** *Assume the SDE in equation 18 has a smooth drift. The adjoint $\mathcal{L}_t^*$ of its infinitesimal generator satisfies the following for any bounded and smooth density $\rho$, where $\mathrm{div}$ is the divergence:*

$$\mathcal{L}_t^* \rho = -\mathrm{div}(\rho \, \Phi(\cdot, t)) + \frac{\gamma(t)^2}{2} \Delta \rho. \tag{51}$$

---

[1] Here we consider an extension of the results in Lee & Chirikjian (2025) to time-dependent drift and diffusion coefficients. Such an extension can be obtained since the results are based on applying the chain rule to a standard form of Stratonovich SDE with respect to only the state variable and hence we can regard $t$ as a constant.

*Proof.* From equation 50, Lemma B.5, and bilinearity of the metric, we get

$$\int_G f\left(\mathcal{L}_t^* \rho\right) \mathrm{d}\mu = \int_G \langle \nabla f, \rho\, \Phi(\cdot, t)\rangle \,\mathrm{d}\mu + \frac{\gamma(t)^2}{2}\int_G \rho\, \Delta f \,\mathrm{d}\mu \tag{52}$$

For the first term, we use the following relationship

$$0 = \int_G \operatorname{div}(f\,\rho\, \Phi(\cdot, t))\,\mathrm{d}\mu$$

$$= \int_G f\, \operatorname{div}(\rho\, \Phi(\cdot, t))\,\mathrm{d}\mu + \int_G \langle \nabla f, \rho\, \Phi(\cdot, t)\rangle\,\mathrm{d}\mu \tag{53}$$

where the first equality follows from the divergence theorem for the compactly supported vector field $f\,\rho\,\Phi(\cdot, t)$, and the second equality follows from the identity $\operatorname{div}(f\,X) = f\operatorname{div}X + \langle \nabla f, X\rangle$ with the vector field $X = \rho\,\Phi(\cdot, t)$ (Chavel, 2006, Theorem III.7.3 and p.150).

For the second term, we use the fact that $\Delta$ is self-adjoint (Chavel, 2006, Theorem III.7.4), that is

$$\int_G \rho\, \Delta f \,\mathrm{d}\mu = \int_G f\, \Delta\rho \,\mathrm{d}\mu \tag{54}$$

Plugging equation 53 and equation 54 into equation 52, we get

$$\int_G f\left(\mathcal{L}_t^* \rho\right) \mathrm{d}\mu = -\int_G f\, \operatorname{div}(\rho\, \Phi(\cdot, t))\,\mathrm{d}\mu + \frac{\gamma(t)^2}{2}\int_G f\, \Delta\rho \,\mathrm{d}\mu \tag{55}$$

Since this holds for every test function $f$, we get equation 51. $\qquad\square$

We are now ready to prove Proposition 4.1. We recall the forward trivialized SDE in equation 9:

$$\mathrm{d}g_t = \mathrm{d}(L_{g_t})_e\left[\gamma(t)\,\mathrm{d}\mathbf{w}_t^{\mathfrak{g}}\right].$$

The proof uses adjoint Kolmogorov forward equation that describes density evolution $\rho_t$ of a stochastic process using the adjoint of its infinitesimal generator (Zhu et al., 2025; Holderrieth et al., 2024):

$$\frac{\partial}{\partial t}\rho_t = \mathcal{L}_t^* \rho_t. \tag{56}$$

**Proposition 4.1** (Reverse trivialized SDE). *If each $p_t(g_t)$ is smooth and positive with respect to the Haar measure, the time-reversal of equation 9 is*

$$\mathrm{d}g_t = \mathrm{d}(L_{g_t})_e\left[-\gamma(t)^2\, \nabla_{\mathfrak{g}}\log p_t(g_t)\,\mathrm{d}t + \gamma(t)\,\mathrm{d}\bar{\mathbf{w}}_t^{\mathfrak{g}}\right], \tag{57}$$

*where $\bar{\mathbf{w}}_t^{\mathfrak{g}}$ is Brownian motion in $\mathfrak{g}$ run backwards in time, and $\nabla_{\mathfrak{g}}\log p_t(g_t)$ is the trivialized score, i.e., the score expressed as an element of the Lie algebra.*

*Proof.* Denoting the density of the forward process by $p_t$ and of the reverse by $\bar{p}_t$, we prove $p_t = \bar{p}_{1-t}$ for all $t \in [0, 1]$ given the initial condition $p_1 = \bar{p}_0$.

From Lemma B.6, the adjoint Kolmogorov forward equation of the forward process is

$$\frac{\partial}{\partial t}p_t = \frac{\gamma(t)^2}{2}\Delta p_t \tag{58}$$

Likewise, the adjoint Kolmogorov forward equation of the proposed reverse process is

$$\frac{\partial}{\partial t}\bar{p}_t = -\gamma(1-t)^2\operatorname{div}(\bar{p}_t\, \nabla\log \bar{p}_t) + \frac{\gamma(1-t)^2}{2}\Delta \bar{p}_t = -\frac{\gamma(1-t)^2}{2}\Delta \bar{p}_t \tag{59}$$

where we used $\operatorname{div}(\bar{p}_t\, \nabla\log \bar{p}_t) = \operatorname{div}(\nabla \bar{p}_t) = \Delta \bar{p}_t$ for the second equality.

Then, the partial derivative of $\bar{p}_{1-t}$ with respect to $t$ is

$$\frac{\partial}{\partial t}\bar{p}_{1-t} = \frac{\gamma(t)^2}{2}\Delta \bar{p}_{1-t} \tag{60}$$

Since this precisely matches the partial derivative of $p_t$ with respect to $t$, given the initial condition $p_1 = \bar{p}_0$ we have that $p_t = \bar{p}_{1-t}$ for all $t \in [0, 1]$, completing the proof. $\qquad\square$

### B.6 Proof of Proposition 4.2

Before proving our main score identity, we state two lemmas regarding conditional random variables on general Lie groups. We distinguish between random variables and their realizations for clarity.

**Lemma B.8.** *Let $G$ be a finite-dimensional Lie group, and let $X, W$ be independent random variables with densities $p_X, p_W$ with respect to the left-invariant measure $\mu$. Let $Y \equiv XW$. Then we have:*

$$p_{Y|X}(y|x) = p_W(x^{-1}y). \tag{61}$$

*Proof.* For any measurable set $A \subseteq G$, we have

$$\mathbb{P}\left(Y \in A | X = x\right) = \mathbb{P}\left(xW \in A\right) = \mathbb{P}\left(W \in x^{-1}A\right) = \int_{x^{-1}A} p_W(w) \,\mathrm{d}\mu(w) \tag{62}$$

where the first equality is due to independence of $X$ and $W$.

Let us apply change of variable $y \equiv xw$. Since $\mathrm{d}\mu(w) = \mathrm{d}\mu(x^{-1}y) = \mathrm{d}\mu(y)$, we have

$$\mathbb{P}\left(Y \in A | X = x\right) = \int_{x^{-1}A} p_W(w) \,\mathrm{d}\mu(w) = \int_A p_W(x^{-1}y) \,\mathrm{d}\mu(y) \tag{63}$$

Since this holds for every measurable set $A \subseteq G$, we get equation 61. $\square$

For the next lemma, we introduce the concept of modular function $\lambda : G \to \mathbb{R}_{>0}$ of a Lie group $G$ that specifies how the left-invariant measure $\mu$ changes under right multiplication. Specifically, for every measurable set $A \subseteq G$ and every $g \in G$, it holds that $\mu(Ag) = \lambda(g)\mu(A)$, and especially we have that $\mathrm{d}\mu(g^{-1}) = \lambda(g^{-1}) \,\mathrm{d}\mu(g) = \lambda(g)^{-1} \,\mathrm{d}\mu(g)$ (Folland, 1994, Section 2.4).

**Lemma B.9.** *Let $G$ be a finite-dimensional Lie group, and let $X, W$ be independent random variables with densities $p_X, p_W$ with respect to $\mu$. Let $Y \equiv XW$. Then we have, with $\lambda$ the modular function:*

$$p_Y(y) = \int_G p_W(w) \, p_X(yw^{-1}) \, \lambda(w)^{-1} \,\mathrm{d}\mu(w). \tag{64}$$

*Proof.* From Lemma B.8, we obtain

$$p_Y(y) = \int_G p_X(x) \, p_{Y|X}(y|x) \,\mathrm{d}\mu(x) = \int_G p_X(x) \, p_W(x^{-1}y) \,\mathrm{d}\mu(x) \tag{65}$$

By applying change of variable $w = x^{-1}y$ with $\mathrm{d}\mu(x) = \mathrm{d}\mu(yw^{-1}) = \mathrm{d}\mu(w^{-1}) = \lambda(w)^{-1}\mathrm{d}\mu(w)$, we get equation 64. $\square$

We are now ready to prove the main score identity.

**Proposition 4.2** (Trivialized target score identity)**.** *For the forward SDE in equation 9, we have*

$$\nabla_{\mathfrak{g}} \log p_t(g_t) = \int_G \nabla_{\mathfrak{g}} \log p_0(g_0) \, p_{0|t}(g_0 \mid g_t) \,\mathrm{d}\mu(g_0) \tag{66}$$

*where the argument of $\log p_0$ is interpreted as $g_t b$ for $b \equiv g_t^{-1} g_0$, and $\nabla_{\mathfrak{g}}$ is taken with respect to $g_t$.*

*Proof.* We note that $g_0$ and $g_t$ as random variables follow a relationship $g_t = g_0 w_t$ where $g_0 \sim p_0$ and $w_t \sim k_t$ are independent. By applying Lemma B.9, we get

$$p_t(g_t) = \int_G k_t(w_t) \, p_0(g_t w_t^{-1}) \, \lambda(w_t)^{-1} \,\mathrm{d}\mu(w_t) \tag{67}$$

By taking trivialized gradient $\nabla_{\mathfrak{g}}$ with respect to the variable $g_t$ and noting that it is a linear operator,

$$\nabla_{\mathfrak{g}} p_t(g_t) = \int_G k_t(w_t) \, \nabla_{\mathfrak{g}} p_0(g_t w_t^{-1}) \, \lambda(w_t)^{-1} \,\mathrm{d}\mu(w_t) \tag{68}$$

Then, using the identity $\nabla_{\mathfrak{g}} p = p \nabla_{\mathfrak{g}} \log p$ from Proposition A.2 we get

$$\nabla_{\mathfrak{g}} \log p_t(g_t) = \int_G \nabla_{\mathfrak{g}} \log p_0(g_t w_t^{-1}) \frac{k_t(w_t)\, p_0(g_t w_t^{-1})}{p_t(g_t)} \lambda(w_t)^{-1} \, \mathrm{d}\mu(w_t) \tag{69}$$

By change of variable $g_0 = g_t w_t^{-1}$ with $\lambda(w_t)^{-1}\, \mathrm{d}\mu(w_t) = \mathrm{d}\mu(w_t^{-1}) = \mathrm{d}\mu(g_t w_t^{-1}) = \mathrm{d}\mu(g_0)$, and using the relationship $p_{t|0}(g_t|g_0) = k_t(g_0^{-1} g_t) = k_t(w_t)$ obtained by Lemma B.8, we get

$$\nabla_{\mathfrak{g}} \log p_t(g_t) = \int_G \nabla_{\mathfrak{g}} \log p_0(g_0) \frac{p_{t|0}(g_t|g_0)\, p_0(g_0)}{p_t(g_t)} \, \mathrm{d}\mu(g_0) \tag{70}$$

where we remark that $\nabla_{\mathfrak{g}}$ is taken with respect to $g_t$, so $\nabla_{\mathfrak{g}} \log p_0(g_0)$ is understood as $\nabla_{\mathfrak{g}} \log p_0(g_t b)$ for $b$ having the value of $g_t^{-1} g_0$. Using Bayes' rule we get equation 66. $\qquad \square$

### B.7    Proof of Proposition 4.3

The proof is inspired by the convolution argument of Akhound-Sadegh et al. (2024), or equivalently an importance sampling estimation applied to target score identity (De Bortoli et al., 2024).

**Proposition 4.3** (Monte Carlo score estimator)**.** *For the forward SDE in equation 9, where $p_0$ is a Boltzmann density specified by a smooth energy $E$, we have*

$$\nabla_{\mathfrak{g}} \log p_t(g_t) = \nabla_{\mathfrak{g}} \log \int_G k_t(w) \exp\bigl(-E(g_t w^{-1}) - \log \lambda(w)\bigr) \mathrm{d}\mu(w) \tag{71}$$

*where $\lambda$ accounts for the change of the Haar measure under inversion, $\mathrm{d}\mu(w^{-1}) = \lambda(w)^{-1} \mathrm{d}\mu(w)$.*

*Proof.* We start at equation 68 and use $\nabla_{\mathfrak{g}} p_t = p_t \nabla_{\mathfrak{g}} \log p_t$ with equation 67 to get

$$\nabla_{\mathfrak{g}} \log p_t(g) = \frac{\int_G k_t(w)\, \nabla_{\mathfrak{g}} p_0(gw^{-1})\, \lambda(w)^{-1}\, \mathrm{d}\mu(w)}{\int_G k_t(w)\, p_0(gw^{-1})\, \lambda(w)^{-1}\, \mathrm{d}\mu(w)} \tag{72}$$

Then using $p_0(\cdot) = e^{-E(\cdot)}/Z$ with the normalization constant $Z$ canceling out, we get

$$\nabla_{\mathfrak{g}} \log p_t(g) = \frac{\int_G k_t(w)\, \nabla_{\mathfrak{g}} e^{-E(g_t w^{-1}) - \log \lambda(w)}\, \mathrm{d}\mu(w)}{\int_G k_t(w)\, e^{-E(g_t w^{-1}) - \log \lambda(w)}\, \mathrm{d}\mu(w)} \tag{73}$$

Using linearity to move $\nabla_{\mathfrak{g}}$ with respect to $g$ out of the integration, and then applying the identity $\nabla_{\mathfrak{g}} f = f \nabla_{\mathfrak{g}} \log f$ with $f$ the denominator, we get equation 71 (by writing $g_t$ instead of $g$). $\qquad \square$

### B.8    Monte Carlo Estimation and Consistency

We discuss sampling estimation of trivialized score based on Proposition 4.3 and prove its consistency. Starting at equation 71 and using Monte Carlo estimation for the integral with $N$ samples, we get:

$$\nabla_{\mathfrak{g}} \log p_t(g) \approx \nabla_{\mathfrak{g}} \log \frac{1}{N} \sum_{i=1}^N \exp\bigl(-E(gw^{(i)-1}) - \log \lambda(w^{(i)})\bigr), \quad w^{(i)} \sim k_t. \tag{74}$$

The $1/N$ factor does not affect the gradient, and hence can be removed. We then obtain an expression of the form $\log \sum_i \exp$ inside the gradient, which allows for a numerically stable evaluation using the LogSumExp trick (Akhound-Sadegh et al., 2024).

We note that the estimator is not unbiased because it has sample mean inside a concave function $\log$. The bias can be understood as the Jensen gap. Nevertheless, we show that it is a consistent estimator.

**Proposition B.12.** *For the forward SDE in equation 9 where $p_0$ is a Boltzmann density specified by a smooth energy $E$, under proper integrability and positivity conditions, equation 74 is a consistent estimator of $\nabla_{\mathfrak{g}} \log p_t$.*

*Proof.* For each $g \in G$, let us denote $h(g, w) \equiv e^{-E(gw^{-1}) - \log \lambda(w)}$ and define

$$J(g) \equiv \int_G k_t(w) \, \nabla_{\mathfrak{g}} h(g, w) \, \mathrm{d}\mu(w), \quad Z(g) \equiv \int_G k_t(w) \, h(g, w) \, \mathrm{d}\mu(w) \tag{75}$$

and define the corresponding empirical means

$$\hat{J}_N(g) \equiv \frac{1}{N} \sum_{i=1}^N \nabla_{\mathfrak{g}} h(g, w^{(i)}), \quad \hat{Z}_N(g) \equiv \frac{1}{N} \sum_{i=1}^N h(g, w^{(i)}), \quad w^{(i)} \sim k_t \tag{76}$$

Then equation 71 and equation 74 can be written respectively as

$$\nabla_{\mathfrak{g}} \log p_t(g) = \nabla_{\mathfrak{g}} \log Z(g) = \frac{J(g)}{Z(g)}, \quad \nabla_{\mathfrak{g}} \log \hat{Z}_N(g) = \frac{\hat{J}_N(g)}{\hat{Z}_N(g)} \tag{77}$$

Assume integrability and positivity conditions

$$\int_G k_t(w) \, \|\nabla_{\mathfrak{g}} h(g, w)\| \, \mathrm{d}\mu(w) < \infty, \quad 0 < \int_G k_t(w) \, h(g, w) \, \mathrm{d}\mu(w) < \infty \tag{78}$$

By the strong law of large numbers

$$\hat{J}_N(g) \xrightarrow{\text{a.s.}} J(g), \quad \hat{Z}_N(g) \xrightarrow{\text{a.s.}} Z(g) \tag{79}$$

Then, by continuous mapping theorem (Shao, 2003, Theorem 1.10 and Example 1.30)

$$\frac{\hat{J}_N(g)}{\hat{Z}_N(g)} \xrightarrow{\text{a.s.}} \frac{J(g)}{Z(g)} \tag{80}$$

thus we have $\nabla_{\mathfrak{g}} \log \hat{Z}_N(g) \xrightarrow{\text{a.s.}} \nabla_{\mathfrak{g}} \log Z(g) = \nabla_{\mathfrak{g}} \log p_t(g)$. $\qquad \square$

### B.9 PRACTICAL IMPLEMENTATION DETAILS

We discuss numerical computation of diffusion sampling $\hat{g}_0 \sim p_0 \propto e^{-E(g)}$ given $E : G \to \mathbb{R}$. The sampling follows a Euler time-discretization (equation 19) of the reverse diffusion (equation 10). For step size $\Delta t > 0$ we use the following for decreasing step indices $m = M, ..., 1$ with $M = 1/\Delta t$:

$$\hat{g}_{m-1} = \hat{g}_m \exp \left( \gamma(m\Delta t)^2 \nabla_{\mathfrak{g}} \log p_{m\Delta t}(\hat{g}_m) \, \Delta t + \gamma(m\Delta t) \, \bar{\mathbf{z}}_m \right), \quad \bar{\mathbf{z}}_m \sim \mathcal{N}(\mathbf{0}, \Delta t \mathbf{I}), \tag{81}$$

$$\hat{g}_M \stackrel{d}{=} \hat{w}_M \sim k_1, \tag{82}$$

where we approximate $k_1 \approx p_1$ for initialization as usually done for variance-exploding diffusion (Song et al., 2020). To run the sampling, we first need to sample from $k_1$ and at each step compute the MC estimator of the score $\nabla_{\mathfrak{g}} \log p_t$ (equation 74). This translates to the following requirements:

1. A method to sample $w \sim k_t$,
2. A method to calculate $\log \lambda(w)$,
3. A method to compute trivialized gradient $\nabla_{\mathfrak{g}}$ of general $f : G \to \mathbb{R}$.

For the first item, we recall the relationship $g_t = g_0 w_t$ with $g_0 \sim p_0$ and $w_t \sim k_t$ independent (proof of Proposition 4.2), and recall the forward process (equation 9). Together, these imply that $w_t \sim k_t$ is described by the following SDE which is identical to the forward SDE but starts at the identity:

$$\mathrm{d}w_t = \mathrm{d}(L_{w_t})_e \left[ \gamma(t) \, \mathrm{d}\mathbf{w}_t^{\mathfrak{g}} \right], \quad w_0 = e. \tag{83}$$

This suggests a natural Euler time-discretization for sampling. For step indices $m = 1, ..., M$ we use:

$$\hat{w}_{m+1} = \hat{w}_m \exp \left( \gamma(m\Delta t) \, \mathbf{z}_m \right), \quad \mathbf{z}_m \sim \mathcal{N}(\mathbf{0}, \Delta t \mathbf{I}), \quad \hat{w}_0 = e, \tag{84}$$

and treat $\hat{w}_m \sim k_{m\Delta t}$.

For the second item, to evaluate the modular function $\lambda(\hat{w}_m)$ for $\hat{w}_m \sim k_{m\Delta t}$ we use the fact that $\hat{w}_m$ can be always written as the following, thanks to the structure of time-discretization (equation 84):

$$\hat{w}_m = \exp(\mathbf{v}_1) \exp(\mathbf{v}_2) \cdots \exp(\mathbf{v}_{m-1}), \tag{85}$$

where each $\mathbf{v}_i = \gamma(i\Delta t)\mathbf{z}_i$ is a Lie algebra element obtained at each sampling step. This allows us to evaluate the modular function only using the knowledge of the Lie bracket $[\cdot, \cdot] : \mathfrak{g} \times \mathfrak{g} \to \mathfrak{g}$:

| Group | Domain | Energy | Noise schedule $(\gamma_{\min}, \gamma_{\max})$ | Step size $\Delta t$ | MC sample size $N$ |
|---|---|---|---|---|---|
| SO(10) | Synthetic | Quadratic | $(0.01, 10)$ | $1/100$ | 100 |
| $\mathrm{Aff}(2, \mathbb{R})$ | Image | VAE | $(0.1, 1)$ | $1/50$ | 4 |
| $\mathrm{Aff}(2, \mathbb{R})$ | Image | Classifier | $(0.1, 1)$ | $1/50$ | 2 |
| $\mathrm{PGL}(3, \mathbb{R})$ | Image | VAE | $(0.01, 0.5)$ | $1/50$ | 4 |
| $\mathrm{PGL}(3, \mathbb{R})$ | Image | Classifier | $(0.01, 0.5)$ | $1/50$ | 2 |
| $\mathrm{SL}(2, \mathbb{R}) \ltimes \mathrm{H}(1, \mathbb{R})$ | PDE | Boundary | $(0.01, 0.5)$ | $1/50$ | 10 |
| $\mathrm{SL}(2, \mathbb{R}) \ltimes (\mathbb{R}^2, +)$ | PDE | Boundary | $(0.01, 0.5)$ | $1/50$ | 10 |

Table 3: Configurations of the TIED sampler used in our experiments.

**Proposition B.13.** *Let $G$ be connected $n$-dimensional Lie group. Under a choice of orthonormal basis $\{\mathbf{e}_1, ..., \mathbf{e}_n\}$, with structure constants $c_{ij}^k$ satisfying $[\mathbf{e}_i, \mathbf{e}_j] = \sum_{k=1}^{n} c_{ij}^k \mathbf{e}_k$, let us denote:*

$$
\mathbf{a} \equiv \left[ \begin{array}{c} \sum_{j=1}^{n} c_{1j}^j \\ \cdots \\ \sum_{j=1}^{n} c_{nj}^j \end{array} \right] \in \mathbb{R}^n, \quad \mathbf{b}_1 \equiv \left[ \begin{array}{c} \langle \mathbf{e}_1, \mathbf{v}_1 \rangle_{\mathfrak{g}} \\ \cdots \\ \langle \mathbf{e}_n, \mathbf{v}_1 \rangle_{\mathfrak{g}} \end{array} \right] \quad \cdots \quad \mathbf{b}_{m-1} \equiv \left[ \begin{array}{c} \langle \mathbf{e}_1, \mathbf{v}_{m-1} \rangle_{\mathfrak{g}} \\ \cdots \\ \langle \mathbf{e}_n, \mathbf{v}_{m-1} \rangle_{\mathfrak{g}} \end{array} \right] \in \mathbb{R}^n.
\tag{86}
$$

*Then the following holds:*

$$
\lambda(\hat{w}_m) = e^{-\mathbf{a}^\top \mathbf{b}_1 - \mathbf{a}^\top \mathbf{b}_2 - \ldots - \mathbf{a}^\top \mathbf{b}_{m-1}}.
\tag{87}
$$

*Proof.* For every connected Lie group $G$, we have (Folland, 1994, Proposition 2.30)

$$
\lambda(g) = \det \mathrm{Ad}(g^{-1})
\tag{88}
$$

where $\mathrm{Ad}(\cdot) : \mathfrak{g} \to \mathfrak{g}$ is the adjoint action of $G$ on the Lie algebra. Using its properties as a linear group representation (Kirillov, 2008, Example 4.8), we get

$$
\lambda(\hat{w}_m) = \frac{1}{\det \mathrm{Ad}(\hat{w}_m)} = \frac{1}{\det \mathrm{Ad}(\exp(\mathbf{v}_1) \cdots \exp(\mathbf{v}_{m-1}))}
$$
$$
= \frac{1}{\det \mathrm{Ad}(\exp(\mathbf{v}_1))} \times \cdots \times \frac{1}{\det \mathrm{Ad}(\exp(\mathbf{v}_{m-1}))}
\tag{89}
$$

Then, using that $\det \mathrm{Ad}(\exp(\mathbf{v})) = \det e^{\mathrm{ad}(\mathbf{v})} = e^{\mathrm{tr}(\mathrm{ad}(\mathbf{v}))}$, where $\mathrm{ad}(\mathbf{v}) : \mathbf{u} \mapsto [\mathbf{v}, \mathbf{u}]$ is a linear map on $\mathfrak{g}$ (Kirillov, 2008, Lemma 3.14) with properties following from bilinearity of the Lie bracket

$$
\mathrm{ad}(\mathbf{v})(\mathbf{e}_i) = [\mathbf{v}, \mathbf{e}_i] = \sum_j \langle \mathbf{e}_j, \mathbf{v} \rangle_{\mathfrak{g}} [\mathbf{e}_j, \mathbf{e}_i] = \sum_j \langle \mathbf{e}_j, \mathbf{v} \rangle_{\mathfrak{g}} \sum_k c_{ji}^k \mathbf{e}_k
\tag{90}
$$

$$
\mathrm{tr}(\mathrm{ad}(\mathbf{v})) = \sum_i \langle \mathbf{e}_i, \mathrm{ad}(\mathbf{v})(\mathbf{e}_i) \rangle_{\mathfrak{g}} = \sum_i \langle \mathbf{e}_i, \mathbf{v} \rangle_{\mathfrak{g}} \sum_j c_{ij}^j
\tag{91}
$$

we obtain equation 87. □

Finally, for the last item, we use Proposition A.1 to directly evaluate the trivialized gradient using automatic differentiation.

We finish the section with a description of the choice of noise schedule $\gamma$ and sampler configurations. We adopt the geometric noise schedule from Song et al. (2020), defined for a choice of $\gamma_{\min} < \gamma_{\max}$ as $\gamma(t) \equiv \gamma_{\min} \cdot (\gamma_{\max}/\gamma_{\min})^t$. It starts at $\gamma(0) = \gamma_{\min}$ and gradually increases to $\gamma(1) = \gamma_{\max}$. We provide the specific configurations of the TIED sampler used in the experiments in Table 3.

# C SUPPLEMENTARY RESULTS

We provide supplementary experimental results that could not be included in the main text due to space constraints. In Table 4, we provide the error bars for MNIST classification test accuracies in Table 1 measured over three repeated tests. In Table 5 and Table 6, we provide the wall-clock runtime of MNIST classification experiment and PDE solving experiment, respectively.

Table 4: MNIST classification test accuracy with std measured over three repeated tests. For FoCal, we were prevented from repeated testing due to high time complexity.

| dataset | MNIST | |
|---|---|---|
| test transformations | none | |
| ResNet18 | 99.35 | |
| test transformations | $\mathrm{Aff}(2, \mathbb{R})$ | $\mathrm{PGL}(3, \mathbb{R})$ |
| affConv / homConv * | 95.08% | 95.71% |
| ResNet18 | 55.48% | 87.95% |
| Energy: VAE evidence lower bound (+ adv. reg.) | | |
| ResNet18 + ITS | $45.79 \pm 0.64\%$ | n/a |
| ResNet18 + FoCal | $86.35 \pm 0.00\%$ | $89.69 \pm 0.00\%$ |
| ResNet18 + Kinetic Langevin | $74.55 \pm 0.17\%$ | $93.27 \pm 0.12\%$ |
| ResNet18 + LieLAC | $94.36 \pm 0.10\%$ | $97.42 \pm 0.06\%$ |
| ResNet18 + TIED (Ours) | $\mathbf{96.84 \pm 0.07\%}$ | $\mathbf{97.45 \pm 0.09\%}$ |
| Energy: Classifier logit confidence | | |
| ResNet18 + ITS | $69.03 \pm 3.58\%$ | n/a |
| ResNet18 + FoCal | $66.73 \pm 0.00\%$ | $86.24 \pm 0.00\%$ |
| ResNet18 + Kinetic Langevin | $53.83 \pm 0.05\ \%$ | $88.26 \pm 0.09\ \%$ |
| ResNet18 + LieLAC | $73.58 \pm 0.34\ \%$ | $85.91 \pm 0.26\ \%$ |
| ResNet18 + TIED (Ours) | $\mathbf{85.21 \pm 0.17\ \%}$ | $\mathbf{89.81 \pm 1.83\ \%}$ |

Table 5: MNIST classification test wall-clock runtime measured on a single NVIDIA A6000 GPU.

| dataset | MNIST | |
|---|---|---|
| test transformations | $\mathrm{Aff}(2, \mathbb{R})$ | $\mathrm{PGL}(3, \mathbb{R})$ |
| affConv / homConv (training time) | 36 hours | 36 hours |
| Energy: VAE evidence lower bound (+ adv. reg.) | | |
| ResNet18 + ITS | 13 mins | n/a |
| ResNet18 + FoCal | 8 hours | 12 hours |
| ResNet18 + Kinetic Langevin | 2 hours | 2 hours |
| ResNet18 + LieLAC | 70 mins | 70 mins |
| ResNet18 + TIED (Ours) | 53 mins | 53 mins |
| Energy: Classifier logit confidence | | |
| ResNet18 + ITS | 13 mins | n/a |
| ResNet18 + FoCal | 2 hours | 2 hours |
| ResNet18 + Kinetic Langevin | 1 hour | 73 mins |
| ResNet18 + LieLAC | 30 mins | 30 mins |
| ResNet18 + TIED (Ours) | 30 mins | 30 mins |

Table 6: PDE solving wall-clock runtime measured on a single NVIDIA 6000 GPU.

| PDE | 1D Heat Eq. | 1D Heat Eq. + Data Aug. | 1D Burgers Eq. |
|---|---|---|---|
| Test transformations | $\mathrm{SL}(2, \mathbb{R}) \ltimes \mathrm{H}(1, \mathbb{R})$ | $\mathrm{SL}(2, \mathbb{R}) \ltimes \mathrm{H}(1, \mathbb{R})$ | $\mathrm{SL}(2, \mathbb{R}) \ltimes (\mathbb{R}^2, +)$ |
| Energy: Distance to training domain | | | |
| DeepONet + FoCal | 17 mins | 17 mins | 7.5 mins |
| DeepONet + Kinetic Langevin | 2 hours | 2 hours | 88 mins |
| DeepONet + LieLAC | 38 secs | 38 secs | 47 secs |
| DeepONet + TIED (Ours) | 43 secs | 43 secs | 47 secs |

