# OpenReview forum: "Inverting Data Transformations via Diffusion Sampling"
_ICLR.cc/2026/Conference — Submitted to ICLR 2026_

### Official Review · Reviewer_EHxJ · 2025-10-27

**Soundness:** 3
**Presentation:** 3
**Contribution:** 3
**Rating:** 8
**Confidence:** 3

**Summary:**

The authors propose an approach that learns to invert Lie symmetries in data via the reverse process of a specialized diffusion process over the group. Specifically, inverse transformations are modeled as a Boltzmann distribution corresponding to an energy term over the data. Scores are recovered from the gradients of the energy functional. Notably, the proposed method can handle non-compact groups that act projectively. Instead of incorporating the proposed approach into existing equivariant models, the authors do the opposite: In applications, the inversion framework is used to robustify non-equivariant models in the presence of challenging perturbations.

**Strengths:**

The authors proposed approach is both novel and creative. The ability to successfully handle non-compact groups and non-linear actions is particularly impressive and important. The paper is also well written.

In particular, I think this paper establishes an important new direction in symmetry discovery, successfully incorporating a modern probabilistic approach consisting of a well-formulated, group-specific version of diffusion and suggesting that meticulously hand-crafted approaches based on engineering equivariant architectures may not be necessary.

**Weaknesses:**

- My main concern involves baselines for comparison. While the main contribution of this paper is clearly techincal, I think it would be useful for the authors to include comparisons with existing equivariant approaches on some of the main experiments for context.

- In addition, several recent works on symmetry discovery are overlooked in the related works, even though these methods handle some of the same tasks (e.g. Homography-perturbed MNIST). One of these methods (Neural Isometries) also handles non-compact and non-linear actions. While these models don't take probabilistic approaches, they are probably worth a short discussion as this paper also represents a contribution to symmetry discovery.

  Latent Space Symmetry Discovery (Yang et al, ICLR 2024) https://arxiv.org/pdf/2310.00105

  Neural Fourier Transform: A General Approach to Equivariant Representation Learning (Koyama et al, ICLR 2024) https://openreview.net/forum?id=eOCvA8iwXH

That said, I expect these concerns to be straight-forward for the authors to address and I recommend acceptance.

  Neural Isometries: Taming Transformations for Equivariant ML (Mitchel et al, NeurIPS 2024) https://arxiv.org/pdf/2405.19296

**Questions:**

Can the authors comment on the scalability of the proposed method? How, if at all, is the complexity tied to the dimensionality of the data?

---

> ### Author Response · Authors · 2025-11-22
>
> We thank the reviewer for their positive appreciation of our work and useful feedback. We are happy that they found our paper well written and that they consider our contributions important for the field.
>
> We have made some presentation and clarity improvements, summarized in the message to all reviewers. We also address the reviewer’s specific comments below.
>
> > My main concern involves baselines for comparison. While the main contribution of this paper is clearly technical, I think it would be useful for the authors to include comparisons with existing equivariant approaches on some of the main experiments for context.
>
> Following the reviewer’s feedback, we have included the performances of the standard affine and homography group convolutional network classifiers (MacDonald et al., 2022) to **Table 1**. We find that our method achieves better accuracy than specialized equivariant architectures while using a generic architecture and without additional training.
>
> The second main experiment tested our approach on notoriously challenging PDE symmetry groups for which, to the best of our knowledge, no method currently exists to build equivariant neural networks. In this experiment, our pretrained models include a neural operator trained with data augmentation.
>
> > Several recent works on symmetry discovery are overlooked in the related works, even though these methods handle some of the same tasks (e.g. Homography-perturbed MNIST). One of these methods (Neural Isometries) also handles non-compact and non-linear actions. While these models don't take probabilistic approaches, they are probably worth a short discussion as this paper also represents a contribution to symmetry discovery.
>
> We thank the reviewer for this comment and for highlighting these works; they are indeed relevant, partially in the sense that we don't have group-specific design components aside from Lie algebra basis and Lie bracket, although a difference is that the discovered symmetry may be approximate in these methods. Applying our method in the context of symmetry discovery would be an interesting direction for future work. We have included a discussion with the mentioned references in the conclusion of the updated manuscript.
>
> > Can the authors comment on the scalability of the proposed method? How, if at all, is the complexity tied to the dimensionality of the data?
>
> This is a great question. An evaluation of the energy at the data is necessary, and therefore, the cost of the model is tied to the cost of a forward pass through the energy-based model. Our experiments have shown that good performance can be achieved with relatively lightweight energy-based models (e.g., the parameter count of the VAE energy models used in image experiments is only 17.4% of the ResNet18 classifier). Otherwise, the sampling method is quite scalable because even though the energy-based model is defined in data space, all gradients and scores are computed with respect to the Lie algebra and are therefore independent of the data dimension. Our method, therefore, performs sampling purely within the group, not in the data space.
>
> To supplement the discussion, we have added wall-clock runtime measurements of the tested methods in **Tables 5 and 6** of the revised manuscript.
>
> Please let us know if we have satisfactorily addressed your concerns, and if you have any other questions, we will be happy to answer them.
>
> ---
> MacDonald et al., Enabling Equivariance for Arbitrary Lie Groups, CVPR 2022.
>
> Yang et al., Latent Space Symmetry Discovery, ICLR 2024.
>
> Koyama et al., Neural Fourier Transform: A General Approach to Equivariant Representation Learning, ICLR 2024.
>
> Mitchel et al., Neural Isometries: Taming Transformations for Equivariant ML, NeurIPS 2024.

---

> > ### Comment · Reviewer_EHxJ · 2025-11-26
> >
> > Thanks for a refreshing rebuttal.
> >
> > Despite the concerns of the other authors, I will maintain my score unless I am convinced otherwise by the input of other reviewers as the discussion progresses.

---

> > > ### Author Response · Authors · 2025-11-27
> > >
> > > Dear reviewer, thank you for your support! We will make sure that the added discussions are clearly represented in the next revision of the paper.

---

### Official Review · Reviewer_Fr8v · 2025-11-02

**Soundness:** 2
**Presentation:** 3
**Contribution:** 3
**Rating:** 4
**Confidence:** 4

**Summary:**

This work introduces Transformation-Inverting Energy Diffusion (TIED) for inverting data transformations. Unlike traditional diffusion models, TIED leverages a novel trivialized target-score identity and enables efficient sampling from the Boltzmann posterior on the group.

TIED effectively addresses the challenges of curved group geometry and rough, multimodal energy landscapes. Specifically, it demonstrates superior performance compared to previous works in two key application scenarios: image homographies and Lie point symmetries for partial differential equations (PDEs).

**Strengths:**

1. The work proposes a novel framework for inverting unknown data transformations from general Lie groups, which breaks through the limitations of traditional diffusion models and provides a new research perspective for the field of transformation inversion.

2. Compared with traditional diffusion models, TIED infers the posterior distribution of transformations based on an energy prior, which significantly accelerates the reconstruction process. Experimental results further confirm that TIED achieves substantial performance improvements over existing methods in targeted tasks.

3. The paper provides a clear and detailed description of the construction process of TIED, accompanied by rigorous mathematical derivations and explanations. This enhances the interpretability and reproducibility of the proposed method.

4. The paper proves a key conclusion: **"Inversion of data transformations can be utilized to achieve test-time equivariance of pre-trained models, without requiring additional modules beyond an approximate energy model (which can even be the predictor itself)"**. This finding expands the application scope of a series of previous works and provides practical guidance for optimizing pre-trained model performance.

**Weaknesses:**

1. The experimental setup lacks sufficient clarity. In Section 5.2, the specific parameters of the affine matrix (a critical factor directly affecting experimental results) are not provided. This omission makes it difficult to fully verify the reliability of the experimental conclusions and hinders the reproducibility of the study.

2. In existing works on affine reconstruction, experimental results are typically presented as quantitative data distributed across the real number axis to reflect result variability. However, this work only reports specific single values, ignoring the importance of repeated experiments. This approach fails to demonstrate the stability and robustness of TIED, weakening the persuasiveness of the experimental results.

3. Some key statements in the paper lack corresponding data support. For example, claims about "reconstructing speed" or "model size" are not backed by specific comparative data , reducing the credibility of the arguments.

**Questions:**

1. Lines 100\~102 state: **"However, since they operate in data space instead of the group, they require large diffusion models and for non-linear actions do not recover transformations that lie on the manifold."** However, the paper does not mention the exact model size of TIED (e.g., number of parameters, computational complexity) or provide a direct comparison with existing methods in terms of model scale. Could the authors supplement this information to verify the advantage of TIED in reducing model size?

2. Lines 361\~363 discuss the difference in the number of steps between the proposed TIED and previous methods. However, the paper does not report the time consumption of both methods during actual operation. Since a smaller number of timesteps does not necessarily translate to higher reconstruction speed (as it may be affected by per-step computational complexity), could the authors provide specific time consumption data and analysis to confirm the efficiency advantage of TIED?

3. Section 5.2, TIED shows considerable performance improvement. For image inverse problems, indicators such as PSNR, SSIM, or FID are commonly used to directly evaluate reconstruction quality. However, this work only uses ResNet as a classifier to assess performance. Given that not all baseline methods are specifically designed for this classification-based evaluation, could the authors explain the rationale for choosing this evaluation metric and supplement it with direct reconstruction quality indicators?

4. What are the specific values of the affine matrix (the ITS achieved an accuracy of 0.89 in the case of pure rotation, and it is well-known that rotation is also a type of affine transformation) in experiment? Were tests conducted under multiple affine transformations? If so, what is the variance of the experiment?

---

> ### Author Response · Authors · 2025-11-22
>
> We thank the reviewer for the helpful feedback and questions. We are happy they have found the experimental evaluation convincing, and the theoretical contributions detailed and rigorous.
>
> Thanks to your feedback and that of the other reviewers, we have significantly improved the paper's clarity and description of experimental setup. All the changes are summarized in the common response and highlighted in the updated manuscript. Below, we address the reviewers' questions in more detail.
>
> > The experimental setup lacks clarity. In Section 5.2, the specific parameters of the affine matrix are not provided. This omission makes it difficult to verify the reliability of the experiments and hinders reproducibility. (...) Were tests conducted under multiple affine transforms?
>
> We thank the reviewer for this feedback and, following it, have significantly improved the description of the experimental setting and moved details previously in the Appendix to the main text to improve clarity. We have also added a new subsection, “Practical Implementation,” that explains how the method is implemented, and an algorithm block. We plan to open-source the code for reproducing the experiments upon acceptance.
>
> Answering the comment on the affine transformation, we use a fixed, 10,000-sized subset of the affNIST dataset (MacDonald et al., 2022; https://www.cs.toronto.edu/~tijmen/affNIST/). The specific amount of rotations, shearing, scaling, and translations are as follows: rotations are randomized uniformly between -20 and +20 degrees; shearing factor is chosen uniformly between -0.2 and +0.2 where a factor of 1 means that a horizontal line turns into a line at 45 degrees; scaling factor is chosen uniformly between 0.8 and 1.2; translations are uniformly chosen under the condition that the digit does not go out of the image boundary.
>
> > In existing works on affine reconstruction, experimental results reflect variability. However, this work only reports single values, ignoring repeated experiments. This fails to demonstrate stability and robustness.
>
> We have taken the reviewers' comments into account and performed the requested experiments across three independent trials to provide more complete results. The results in the updated manuscript (**Table 4**) show that our method is stable and robust.
>
> > Claims about "reconstructing speed" or "model size" are not backed by specific data. (...) Lines 100~102: "However, since they operate in data space instead of the group, they require large diffusion models (...)" However, the paper does not mention the model size of TIED (e.g., number of parameters, computational cost) or provide a comparison with existing methods. Could the authors supplement this information? (...) Could the authors provide specific time consumption data and analysis to confirm the efficiency advantage?
>
> We have expanded the discussion of reconstruction (sampling) speed and model size in the updated manuscript in response to the reviewer’s feedback. The runtime measurements have been added to **Tables 5 and 6**, and the speed of TIED is comparable or better than the baselines. For the model size, we remark that we don’t train any score networks; for MNIST, we rely on pretrained energy-based models, which is either the ResNet18 classifier itself, or a VAE which has only 17.4% of the parameters of the classifier; and for the rest the experiments we rely on no additional networks at all. In each experiment, we use the identical energy function as the baselines.
>
> > For image inverse problems, indicators such as PSNR, SSIM, or FID are commonly used to directly evaluate reconstruction quality. However, this work only uses ResNet as a classifier to assess performance. (...) Could the authors explain the rationale for choosing this evaluation metric and supplement it with direct reconstruction quality indicators?
>
> Since our key application is test-time equivariance for robust predictions with pretrained models, measuring the end-to-end accuracy of the pretrained model under an unknown transformation is a correct evaluation metric to assess the usefulness of an inverse transformation method. Following the reviewer’s suggestion, we ran a supplementary measurement of FID in the image space, measuring the discrepancy from the set of non-transformed images (specifically, the training set of the ResNet18 classifier), and added the results in **Table 1**. The results show that the end-to-end accuracy and FID measurements overall agree well, supplementing the utility of our method. We also considered PSNR and SSIM metrics, but they were not applicable to our test datasets, which are taken from MacDonald et al. (2022), that do not provide pairs of clean and transformed images.
>
> Please let us know if we have satisfactorily addressed your concerns, and if you have any other questions, we will be happy to answer them.
>
> ---
> MacDonald et al., Enabling equivariance for arbitrary Lie groups, CVPR 2022.

---

> ### Author Response · Authors · 2025-11-27
>
> Dear reviewer, thank you again for taking the time to review our work.
>
> As the end of the discussion phase is approaching (Dec 2), we would like to kindly ask if our rebuttal has addressed the raised concerns. If there are any remaining concerns, we would be happy to respond to them.

---

### Official Review · Reviewer_8yTY · 2025-11-02

**Soundness:** 3
**Presentation:** 1
**Contribution:** 2
**Rating:** 4
**Confidence:** 2

**Summary:**

The paper “Inverting Data Transformations via Diffusion Sampling” introduces TIED (Transformation-Inverting Energy Diffusion), a probabilistic framework for recovering unknown transformations—such as rotations, perspective warps, or coordinate shifts—that distort data. It models the inverse transformation as a Boltzmann distribution over a Lie group, where low-energy transformations correspond to more “natural” data. TIED performs diffusion sampling directly on the transformation manifold, combining precise gradient-based local optimization with stochastic exploration to efficiently find likely inverse transformations.

The experiments demonstrate that TIED performs favorably compared to gradient-based and optimization baselines across three settings:
(1) synthetic high-dimensional sampling tasks,
(2) image classification under random affine and homography distortions, and
(3) partial differential equation (PDE) problems with hidden symmetry transformations.

**Strengths:**

- Overall, I found the setup both novel and conceptually interesting. The method enables density estimation on Lie groups using purely data-driven constraints, bridging geometry and probabilistic modeling elegantly.
- The experiments show that applying TIED before inference improves neural network accuracy and consistency under unseen transformations, providing a convincing proof of concept for the proposed idea.

**Weaknesses:**

- Despite its technical novelty, the paper suffers from weak presentation and clarity. The writing is often dense, and key visual aids (e.g., Figure 1) are underexplained—either the caption needs more detail or the figure itself should be redesigned to convey the main idea clearly.
- Conceptually, the work frames “canonicalization” as a way to achieve equivariance through energy-based sampling on Lie groups. However, the paper lacks a clear discussion of how this approach fits within the broader spectrum of data augmentation versus architectural equivariance methods.
- The experimental evaluation, while promising, remains proof-of-concept rather than comprehensive. The authors convincingly demonstrate that the method functions as intended, but a more thorough benchmarking—especially against equivariant neural networks—would be essential to assess its broader impact.

**Questions:**

Overall, this appears like a creative and well-motivated idea with some good arguments with theoretical underpinnings and good initial results. However, the paper currently reads more as a conceptual demonstration than a fully mature study. I would not recommend it for acceptance in its current form, but with clearer exposition and stronger experimental validation, it could become a valuable contribution.

---

> ### Author Response · Authors · 2025-11-22
>
> We thank the reviewer for their constructive comments. We appreciate that they have found our setup and method novel, creative and well-motivated.
>
> > Despite its technical novelty, the paper suffers from weak presentation and clarity. The writing is often dense, and key visual aids (e.g., Figure 1) are underexplained—either the caption needs more detail or the figure itself should be redesigned to convey the main idea clearly.
>
> Thanks to your feedback and that of the other reviewers, we have significantly improved the presentation and writing of the paper. All the changes are summarized in the common response and highlighted in the updated manuscript. This includes significant details added to the captions of Figures 1 to 3.
>
> To specifically clarify Figure 1, it describes the problem setup and method using a probabilistic graphical model. The design is inspired by the common graphical model interpretation of diffusion models (Ho et al., 2020; Song et al., 2021). The observed variables are in gray and the unobserved ones in white. We have significantly expanded the explanations in the caption.
>
> > Conceptually, the work frames “canonicalization” as a way to achieve equivariance through energy-based sampling on Lie groups. However, the paper lacks a clear discussion of how this approach fits within the broader spectrum of data augmentation versus architectural equivariance methods.
>
> Thank you for the constructive feedback. We have added a discussion of data augmentation and equivariance methods in **Section 2.2**. Canonicalization approaches offer an alternative, test-time paradigm to equivariant neural networks that require specialized architectures, and also to (training-time) data augmentation that require more expensive training procedures.
>
> > The experimental evaluation, while promising, remains proof-of-concept rather than comprehensive. The authors convincingly demonstrate that the method functions as intended, but a more thorough benchmarking—especially against equivariant neural networks—would be essential to assess its broader impact. (...) The paper currently reads more as a conceptual demonstration than a fully mature study. I would not recommend it for acceptance in its current form, but with clearer exposition and stronger experimental validation, it could become a valuable contribution.
>
> Following the reviewer’s feedback, we have included the performances of the standard affine and homography group convolutional network classifiers from MacDonald et al., (2022) to **Table 1**. Our approach surpasses these specialized equivariant architectures while using a generic architecture and no additional training or fine-tuning. Furthermore, we added to **Table 1** an alternative evaluation metric (FID) that measures the quality of inverse-transformed images, which supplements our benchmarking. In addition, we included wall-clock runtime measurements of the tested methods in **Tables 5 and 6** to support the practicality of the method.
>
> We would like to remark that we have also tested our approach on notoriously challenging PDE symmetry groups and benchmarked against very recent state-of-the-art methods such as FoCal and LieLAC. For these PDE symmetry groups, to the best of our knowledge, there is currently no method for building equivariant neural networks. Therefore, we believe our benchmarking within the currently available methods is comprehensive.
>
> Finally, we wish to highlight that some of the important contributions of our work are theoretical. This work introduces, among other things, the first diffusion sampler for general Lie groups, which we believe could be very useful for applications such as generative modelling and solving inverse problems on groups.
>
> Please let us know if we have satisfactorily addressed your concerns, and if you have any other questions, we will be happy to answer them.
>
> ---
>
> Ho et al., Denoising diffusion probabilistic models, NeurIPS 2020.
>
> Song et al., Score-based generative modeling through stochastic differential equations, ICLR 2021.
>
> MacDonald et al., Enabling equivariance for arbitrary Lie groups, CVPR 2022.

---

> ### Author Response · Authors · 2025-11-27
>
> Dear reviewer, thank you again for taking the time to review our work.
>
> As the end of the discussion phase is approaching (Dec 2), we would like to kindly ask if our rebuttal has addressed the raised concerns. If there are any remaining concerns, we would be happy to respond to them.

---

### Official Review · Reviewer_Pfbk · 2025-11-04

**Soundness:** 3
**Presentation:** 2
**Contribution:** 4
**Rating:** 4
**Confidence:** 4

**Summary:**

This paper presents TIED, which is a method that can de-transform an image from an arbitrary group action to invert its data transformation. This is a novel idea and in fact learning this inverse transformation process is valid and interpretable and robust. The authors also proposed a test-time equivariance algorithm which helps pretrained networks to be G-equivariant. The experiments are performed on image recognition and PDE data.

**Strengths:**

This paper presents a very interesting and working algorithm.

The problem of group-equivariant learning has been studied for a long time. As an important property of physics (symmetry), the learning/prediction outcomes should be equivariant to group transformations. However, most of the prior methods (i) data augmentation (ii) learning a standard position or (iii) group convolutions can be (i) not robust (ii) hard to define (iii) expensive. This framework nicely combines the strength of the diffusion model to learn the inverse process, which seems to provide a good solution to this community.

**Weaknesses:**

The main weakness of this paper is the presentation. Some figures and experimental results are hard to understand easily (even after a couple of re-reads from the reviewer's side - who is familiar with the field).

Please see the questions below.

**Questions:**

1. Could the authors explain Figure 2? The reviewer would request the authors to talk about t, sigma, what are the blue and green lines. Are they the same thing? (one is log density and the other is the original density)? What does the author mean by "Note the multimodal nature of the posterior, with modes centered around g = −20◦, 135◦, 180◦." Does this come from the available dataset? For an image network, which image network is this?

2. Figure 3 requires a much better description, as this is a very interesting experiment. First, it is not immediately understandable what X_{1,1}^2 represents. And what is SO(10)? In fact SO(2), SO(3) also need to be defined. And what does it mean by dim G = 45? And how many samples kinetic Langevin algorithm was used? Why is mean of X_{1,1} important? The reviewer believes the readers will apprecaite this experiment more if the authors can write a better description.

3. In Table 1, why ResNet18+ITS may cause n/a? Would the authors try to center the words like "energy: VAE evidence lower bound (+ adv. reg.)" or make these fonts bold (to better separate these lines)?

4. Would the authors comment more on the computational requirement? Since it may implicitly require a lot of computation to solve the inverse process. With the same computational budget, one might already be able to get a pretty well generalized model (from pre-training with Lie-equivariant learning methods). In case the reviewer misses the context, would a trained Lie inverse transformation model be able to reverse both the PDE and images once it is trained? To be clear as well, the result might not be able to generalize to images that haven't been seen before right (since the energe function will be wrong)?

The reviewer is very likely to improve the rating if one (or more) of the prior questions can answered by the authors.

---

> ### Author Response · Authors · 2025-11-22
>
> We thank the reviewer for their positive comments on the soundness and significance of our work. We are glad they found our method interesting and that it provided a good working solution to the problem of interest.
>
> > The main weakness is the presentation. Some figures and experimental results are hard to understand easily.
>
> Thanks to your feedback and that of the other reviewers, we have significantly improved the presentation and writing. All the changes are summarized in the common response and highlighted in the updated manuscript.
>
> > Could the authors explain Figure 2?
>
> Figure 2 has two goals. The first one is to visualize the energy (blue, top left) and probability density (green, bottom left) associated with the LogSumExp energy of a ResNet classifier for different transforms of a data $\tilde{x}$. The energy quantifies how confident the classifier is in its predictions and is low when the data is in-distribution. We find that there are three transforms (orientations) that have low energy because the 8-digit has many plausible orientations.
>
> The second goal is to show visually that the density associated with the energy (bottom left) is spiky and hard to sample from. However, using a diffusion process, we can obtain “noisy” densities that are smoother. Our sampler works by estimating the scores along these noisy distributions. $t$ indicates the evolution of the density along the forward process. Note that this is the argument outlined by Song & Ermon 2019 to motivate the use of diffusion models.
>
> In the updated manuscript, we have simplified the design of the figure, explicitly labelling the energy and removing the $\sigma$ labels (associated with the noise schedule, which could be confusing). We have expanded the caption to explain the figure better and specify which network is used to obtain the energy.
>
> > Figure 3 requires a better description, as this is an interesting experiment.
>
> We agree that this is an interesting experiment; we have therefore clarified the setup in the manuscript. SO(10) is the group of 10×10 orthogonal matrices $X$ with determinant 1, similar to how SO(3) is defined, and $X_{1,1}$ is the top-left element of the matrix $X$. The dimension of the Lie group (number of free parameters of the 10 x 10 matrix) is 45. We have chosen this group as its large dimension offers a much more challenging sampling testbed than groups like SO(3) and thus shows the effectiveness of our method.
>
> For kinetic Langevin, we used 100 samples in parallel for 100k steps to ensure convergence.
>
> The element $X_{1,1}$ is important as it allows us to visualize the validity and convergence of the samplers as in Figure 3. As the energy is $E(X) = -10 X_{1,1}^2$, we see that the true distribution of $X_{1,1}$ must have two symmetric modes around zero, and as the sampling converges, the mean of $X_{1,1}$ must converge to zero due to the symmetry. These are indeed demonstrated in the two panels of Figure 3.
>
> > In Table 1, why is ResNet18+ITS n/a?
>
> The reason for the n/a for ResNet+ITS for the projective group is that ITS is designed explicitly for affine transforms, and it is not clear how to expand it to projective transforms. We have clarified this in the updated manuscript.
>
> > Would the authors comment on the computational cost? (...) With the same budget, one might already be able to get a well generalized model from equivariant learning.
>
> In practice, diffusion sampling can be done with a modest budget. For example, testing for 10k affine/homography transformed 40x40 MNIST images takes 1-2 hours on an RTX A6000 GPU. For comparison, training an affine/homography-invariant CNN classifier proposed in MacDonald et al. (2022) takes 20 hours on the same GPU using the reported hyperparameters. We have clarified this in the **Appendix C** of the updated manuscript.
>
> > Would a trained transformation model be able to reverse both PDEs and images? The result might not be able to generalize to images not been seen before (as the energy will be wrong)?
>
> We would like to remark that our method is training-free. Instead of training a separate transformation model such as a score network, we directly estimate the score at test time using the energy.
>
> We do however, assume a suitable choice of the energy. While we expect it to be domain-specific to some degree (we use different energies for images and PDEs), we find it possible for an energy to generalize to unseen transforms. The classifier energy we used for MNIST is computed with an ResNet18 classifier that has only seen “clean” MNIST images. Still, it generalizes meaningfully to both affine/homography transforms at test time under our method.
>
> Please let us know if we have satisfactorily addressed your concerns, and if you have any other questions, we will be happy to answer them.
>
> ---
> Song & Ermon, Generative modeling by estimating gradients of the data distribution, NeurIPS 2019.
>
> MacDonald et al., Enabling Equivariance for Arbitrary Lie Groups, CVPR 2022.

---

> ### Author Response · Authors · 2025-11-27
>
> Dear reviewer, thank you again for taking the time to review our work.
>
> As the end of the discussion phase is approaching (Dec 2), we would like to kindly ask if our rebuttal has addressed the raised concerns. If there are any remaining concerns, we would be happy to respond to them.

---

### Author Response · Authors · 2025-11-22
**Response to all reviewers**

We thank the reviewers for their constructive feedback that has helped us significantly improve the paper, especially on the presentation side.

We have clarified the core aspects of our methods and contributions. In summary, they consist of the following:
- We propose a **training-free** method for transformation inversion using simple energy-based diffusion. The energy function quantifies the likelihood of different transformations of a sample. Using our method, we build a **test-time equivariance pipeline** for non-equivariant pretrained models.
- This has significant advantages. First, **the method is compatible with any pretrained predictor and any energy**. The energy can also be a predefined function built from domain expertise or even obtained from the predictor $f$ itself. **We therefore do not train a diffusion model.**
- Second, even if training a specialized equivariant architecture was considered instead of our purely test-time approach, **we address general groups for which there is no known method to build equivariant networks**. Our method is tested and successful for equivariance to very challenging groups, such as point symmetry groups of PDEs.
- Finally, sampling from rough energy functions is generally a challenging problem, and doing that on groups is even more so. It has recently been found that diffusion sampling (which is distinct but related to diffusion models) is a very efficient method for fast sampling from energy functions. **Our results extend, for the first time, diffusion sampling to general Lie groups (without assumptions of an Abelian group or a bi-invariant metric) and could be repurposed for other applications.**

Our work, therefore, bridges geometric deep learning and probabilistic inference/sampling via diffusion in a novel and rigorous way.

We summarize the changes we have made to improve the presentation and clarity of the paper (making use of the additional page), following feedback from all reviewers.
- Revised introduction to more clearly convey the goals and method early in the paper.
- Addition of significant details to the captions of Figures 1, 2 and 3 and revision of the design of Figure 2.
- Significant rewriting of Section 4, presenting theoretical results on diffusion sampling with trivialization.
  - We have tried to make this section less dense and more understandable by moving some of the technicalities to the appendix and introducing the core concepts in a more accessible way.
  - We have added a new subsection, “Practical implementation”, which includes an algorithm, to help better convey the overall procedure.
- Expanded and revised Appendix. The new Appendix includes an additional background section that explains Lie groups, tangent maps, gradients on Lie groups, and trivialization in more detail. In addition, we have revised and simplified the presentation of the results and proofs. Intermediate lemmas and propositions should be readily reusable by future works.
- Improved description of the experimental setup, including details needed for reproducibility.

We have also provided additional experimental results for a more comprehensive validation of the method.
  - Supplemented the affine/homography MNIST experiment by adding an equivariant neural network baseline, and adding an FID evaluation metric.
  - Added error bars from repeated experiments, supporting the stability and robustness of the method.
  - Supplemented efficiency demonstrations by adding wall-clock runtime measurements for all tested methods.

The revisions in the updated manuscript are highlighted in blue.

---

### Author Response · Authors · 2025-11-30
**Summary overview of submission 22862**

Dear AC,

We appreciate your efforts in this challenging situation.

To assist with assessing our work, we would like to provide an overview:

---

Contributions
- We tackle the problem of **transformation inversion**, where data is distorted by an unknown Lie-group transformation and the goal is to find an inverse in probability. A key application is **test-time equivariance** for non-equivariant pretrained models. We frame it as a blind inverse problem and consider the Bayesian posterior solution.
- We propose an **energy-based formulation** for transformation inversion. The energy quantifies the likelihood of different transformations of the data, and its choice is flexible.
- We propose TIED, a **training-free method** for transformation inversion using **energy-based diffusion on Lie groups**. TIED only requires an energy model and knowledge of Lie algebra, and handles multi-modal and rugged energies. **It is the first extension of diffusion sampling to general Lie groups and nonlinear actions.**
- In a sampling problem on a high-dimensional Lie group SO(10), homography and affine-invariant image classification with a pretrained ResNet, and neural PDE solving under unseen initial conditions via point symmetries, TIED **outperforms strong optimization and sampling baselines.**

---

Positive Assessments
- **Reviewer Pfbk:** “a very interesting and working algorithm.” “nicely combines the strength of the diffusion model (...) a good solution to this community.” **“The reviewer is very likely to improve the rating if one (or more) of the prior questions can be answered.”**
- **Reviewer 8yTY:** “I found the setup both novel and conceptually interesting. (...) bridging geometry and probabilistic modeling elegantly.” “The authors convincingly demonstrate that the method functions as intended…” “a creative and well-motivated idea with some good arguments with theoretical underpinnings and good initial results.”
- **Reviewer Fr8v:** “a novel framework for inverting data transformations (...) provides a new research perspective...” “Experimental results further confirm that TIED achieves substantial performance improvements...” “clear and detailed description of the construction (...) accompanied by rigorous mathematical derivations and explanations.”
- **Reviewer EHxJ:** “novel and creative. The ability to successfully handle non-compact groups and non-linear actions is particularly impressive and important. The paper is also well written.” “establishes an important new direction in symmetry discovery…” **“I expect these concerns to be straight-forward for the authors to address and I recommend acceptance.”**

---

Rebuttal
- **Reviewer Pfbk:** Significantly improved presentation and writing (common response), including presentations of Figures 2 and 3, description of synthetic experiment (Section 5.1), and clarification of n/a in Table 1. Clarified modest computational cost via runtime measurements including equivariant networks in Appendix C. Clarified generalizability of the energy function to unseen transformations with evidence from image experiments.
- **Reviewer 8yTY:** Significantly improved presentation and writing (common response), including presentations of Figures 1-3, methods in Section 4, and discussion of data augmentation and equivariant networks in Section 2.2. Made empirical validations stronger by adding **(a)** Equivariant network baselines to image experiments, while clarifying that such baselines currently do not exist for the considered PDEs, **(b)** FID metric in Table 1, **(c)** Error bars in Tables 2 and 4, and **(d)** Wall-clock runtimes in Tables 5 and 6.
- **Reviewer Fr8v:** Improved clarity of experimental setup (Section 4.3 to 5, Appendix B.9), and clarified the details of affine transformations. Demonstrated stability and robustness via error bars (Tables 2 and 4). Clarified practical speed via wall-clock runtimes in Table 5 and 6. Clarified model sizes by remarking that we do not train score networks, and by providing evidence that energy functions can be pretrained and lightweight. Clarified the validity of classification accuracy metric, and added FID metric in Table 1.
- **Reviewer EHxJ:** Added equivariant network baselines to image experiments, while clarifying that such baselines currently do not exist for the considered PDEs. Added discussion of symmetry discovery prior works in Conclusion. Clarified scalability of the method by discussing dimensionality, adding wall-clock runtimes in Table 5 and 6, and providing evidence that energy functions can be lightweight.

All reviewers recognized the novelty, clear motivations, technical and empirical soundness, and potential impact.

We hope this summary is helpful for your assessment.

Kind regards,

Authors of submission 22862

---

### Meta-Review · Area_Chair_mRE8 · 2026-01-12

**Summary:**

readability and more than proof-of-concept experiments (8yTY), readability and compute (Pfbk), weakness of experiments (Fr8v, EHxJ)

**Reviewer Concerns:**

Authors did provide additional experiments, however most of the updated material in the paper is analytical

**Reviewer Scores:**

It seems likely the additional experiments would not have helped a lot to cross the "borderline-" region where the paper currently stands.

---

### Decision · Program_Chairs · 2026-01-26

Reject